# Identification of orphan ligand-receptor relationships using a cell-based CRISPRa enrichment screening platform

Dirk H Siepe[1], Lukas T Henneberg[1], Steven C Wilson[1], Gaelen T Hess[2], Michael C Bassik[2], Kai Zinn[3], K Christopher Garcia[1,4,5]*

[1]Department of Molecular and Cellular Physiology, Stanford University, Stanford, United States; [2]Stanford ChEM-H, Department of Genetics, Stanford University, Stanford, United States; [3]Division of Biology and Biological Engineering, California Institute of Technology, Pasadena, United States; [4]Howard Hughes Medical Institute, Stanford University School of Medicine, Stanford, United States; [5]Department of Structural Biology, Stanford University School of Medicine, Stanford, United States

*For correspondence:
kcgarcia@stanford.edu

Competing interest: The authors declare that no competing interests exist.

**Abstract** Secreted proteins, which include cytokines, hormones, and growth factors, are extra-cellular ligands that control key signaling pathways mediating cell-cell communication within and between tissues and organs. Many drugs target secreted ligands and their cell surface receptors. Still, there are hundreds of secreted human proteins that either have no identified receptors ('orphans') or are likely to act through cell surface receptors that have not yet been characterized. Discovery of secreted ligand-receptor interactions by high-throughput screening has been problematic, because the most commonly used high-throughput methods for protein-protein interaction (PPI) screening are not optimized for extracellular interactions. Cell-based screening is a promising technology for the deorphanization of ligand-receptor interactions, because multimerized ligands can enrich for cells expressing low affinity cell surface receptors, and such methods do not require purification of receptor extracellular domains. Here, we present a proteo-genomic cell-based CRISPR activation (CRISPRa) enrichment screening platform employing customized pooled cell surface receptor sgRNA libraries in combination with a magnetic bead selection-based enrichment workflow for rapid, parallel ligand-receptor deorphanization. We curated 80 potentially high-value orphan secreted proteins and ultimately screened 20 secreted ligands against two cell sgRNA libraries with targeted expression of all single-pass (TM1) or multi-pass transmembrane (TM2+) receptors by CRISPRa. We identified previously unknown interactions in 12 of these screens, and validated several of them using surface plasmon resonance and/or cell binding assays. The newly deorphanized ligands include three receptor protein tyrosine phosphatase (RPTP) ligands and a chemokine-like protein that binds to killer immunoglobulin-like receptors (KIRs). These new interactions provide a resource for future investigations of interactions between the human-secreted and membrane proteomes.

## Editor's evaluation

This paper reports the development and application of a proteo-genomic screening platform to identify protein-protein interactions between secreted proteins and their cell surface receptors. The authors use a CRISPRa-based approach to overexpress membrane proteins in cells and then use magnetic cell sorting to identify receptors that bind candidate ligands. This approach led to the identification of several novel interaction pairs that were then validated biochemically, including receptor tyrosine phosphatase ligands and other interactions with implications for immune system function. The work is likely to be relevant to a wide variety of fields including biochemistry and signal transduction research.

## Introduction

The human proteome can be envisioned as an array of nodes grouped into local communities, where each node represents one protein and each local community represents a protein complex or network (*Budayeva and Kirkpatrick, 2020*; *Huttlin et al., 2017*). These communities determine physiological function and subcellular localization. Many communities include secreted protein ligands, their cell surface receptors, and signaling molecules that bind to the receptors. The human secretome on its own constitutes approximately 15% of all human genes and encodes more than 4000 different proteins (*Uhlén et al., 2019*) with a wide range of tissue expression (*Figure 1A and B*). Most of the new drugs developed in recent years target secreted proteins and their receptors, and new therapeutic targets are likely to emerge from screens to identify ligand-receptor interactions (*Clark et al., 2003*; *Stastna and Van Eyk, 2012*).

Mapping of interactions that occur at the cell surface has significantly lagged behind that of intracellular interactions, because the most widely used high-throughput protein-protein interaction (PPI) screening methods, including affinity purification/mass spectrometry (AP/MS), yeast two-hybrid screening (Y2H), and phage display, are not well suited to analysis of extracellular domain (ECD) interactions (*Havugimana et al., 2012*; *Huttlin et al., 2015*; *Krogan et al., 2006*; *Martinez-Martin, 2017*). ECD interactions are often of low affinity, with $K_D$s in the micromolar range, and can have fast dissociation rates, rendering them difficult to detect since they may not produce stable complexes (*Honig and Shapiro, 2020*). As a consequence, ECD interactions are generally underrepresented in screens that rely on the formation of such complexes (*Braun et al., 2009*; *Martinez-Martin et al., 2019*; *Özkan et al., 2013*; *Söllner and Wright, 2009*; *Wojtowicz et al., 2020*). In addition, many putative ECD interactions reported by AP/MS and Y2H protein interaction databases have the tendency to be false positives. AP/MS produces false positives for cell surface proteins due to incomplete solubilization of membranes, leading to identification of indirect interactions. Y2H examines interactions inside the cell, but most ECDs have disulfide bonds and glycosylation sites. To acquire these modifications and fold correctly, cell surface and secreted proteins must move through the secretory pathway. Because of this, ECD interactions detected by Y2H are often false positives due to domain misfolding. Similar issues apply to phage display and to microarrays in which mRNAs are translated on a chip. Thus, while these high-throughput methods can identify interactions with the cytoplasmic domains of receptors, they usually fail to find genuine ECD interactions.

Successful high-throughput screens to detect weak ECD interactions in vitro have taken advantage of avidity effects by expressing ECDs as fusions with multimerization domains. In such binary interaction screens, one protein (the bait) is applied to a surface, and the other (the prey) is in solution. Prey binding to the bait is assessed using colorimetric or fluorescent detection. These methods include AVEXIS, ECIA, apECIA, alpha-Screen, and BPIA, which are carried out using ELISA plates, chips, or beads (*Braun et al., 2009*; *Bushell et al., 2008*; *Li et al., 2017*; *Martinez-Martin, 2017*; *Shilts et al., 2022*; *Taouji et al., 2009*). However, in vitro screens have limitations. They require robotic high-throughput instrumentation and are time-consuming and expensive to carry out on a large scale, since they require synthesis of ECD coding regions and expression of individual bait and prey proteins. In addition, in vitro screens cannot usually assess binding to ECDs of receptors that span the membrane multiple times, because such ECDs are often composed of noncontiguous loops and cannot be easily expressed in a soluble form. Furthermore, in vitro binary interaction mapping technologies lack the natural spatial context of the cell membrane. They may also fail in cases where cofactors and/or post-translational modifications are required for binding.

To address these issues, several groups have successfully developed cell-based screens for phenotypical screens (*Bassik et al., 2013*; *Han et al., 2020*; *Kamber et al., 2021*), uncovering signaling cascades (*Breslow et al., 2018*; *Wisnovsky et al., 2021*) or to study interactions between cell surface receptors using receptor ECDs (*Chong et al., 2018*), that take advantage of CRISPR technology (*Cong et al., 2013*; *Jinek et al., 2013*; *Mali et al., 2013*). In CRISPR activation (CRISPRa) screens such as the one described here, gene expression is induced by targeting transcriptional activators (*Chavez et al., 2016*; *Chong et al., 2018*; *Tycko et al., 2017*) to their control elements using sgRNAs (*Kampmann, 2018*; *Morgens et al., 2016*; *Tanenbaum et al., 2014*). Utilizing CRISPRa pooled sgRNA libraries eliminates the need to create expensive collection of synthetic genes, and in addition allows a forward positive screening workflow which enables a higher dynamic range compared to loss-of-function

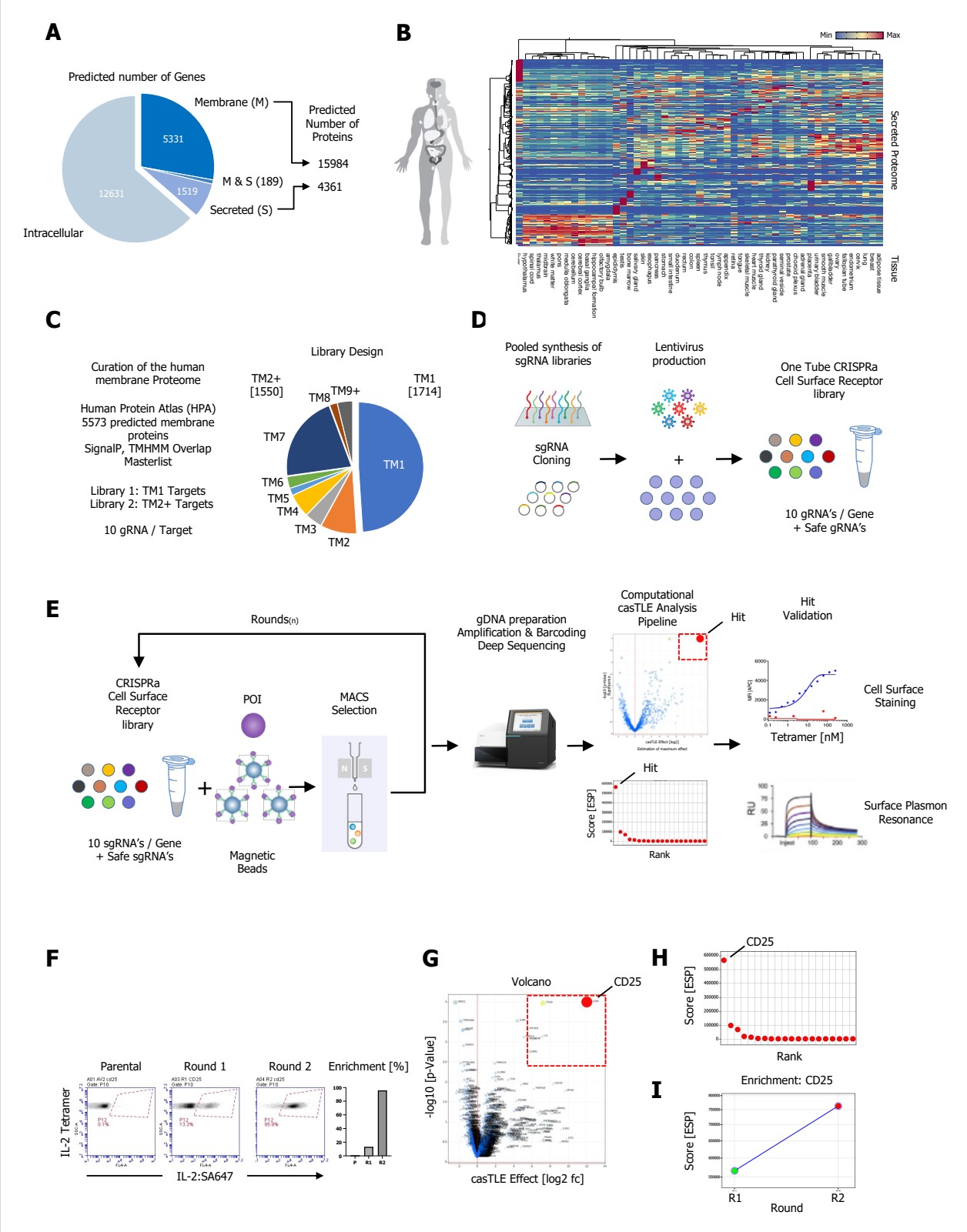

**Figure 1.** A CRISPR activating enrichment screening platform. Curation of the human membrane proteome, cell surface library design, validation, and benchmark screen. (**A**) Human membrane and secreted proteome; left panel: predicted number of intracellular, membrane (**M**), and secreted (**S**) genes, with a total number of approximately 5520 human protein-coding genes predicted to encode ~15,984 membrane-spanning proteins including mapped, alternative splice variants and isoforms. (**B**) Secreted proteome visualized by two-way hierarchical clustering of normalized mRNA expression data from

*Figure 1 continued*

normal tissue. (**C**) Human membrane proteome curation and workflow of the cell surface library design. (**D**) Pooled, customized, and target-specific single-pass transmembrane (TM1) and multi-pass transmembrane (TM2+) sgRNA libraries (10 sgRNA/target) were designed, cloned, and lentivirally infected into K562-SunCas9 cells at low multiplicity of infection (MOI). (**E**) Schematic overview of the CRISPR activation (CRISPRa) enrichment screening platform. A protein of interest (POI) is complexed with magnetic beads and screened against customized CRISPRa cell surface receptor library, followed by consecutive rounds of magnetic-activated cell sorting (MACS) positive selection. In the final step, genomic DNA is extracted from the selected, target enriched library round(s), barcoded, subjected to deep sequencing and analyzed using the casTLE statistical framework to identify potential hits. CRISPRa hits are then subjected to various orthogonal validation methods. (**F–I**) Benchmark CRISPRa enrichment screen using human IL-2, performing two consecutive rounds of magnetic bead selection followed by gDNA extraction, barcoding, and deep sequencing. (**F**) Enrichment over two rounds of consecutive magnetic bead selection by tetramer staining with human IL-2 post selection (Parental, Round 1, and Round 2). (**G**) Visualization of the deep sequencing analysis. Results are visualized by x/y scatter plot: casTLE-Score (log2); pValue (−log10), size of the hit represents the casTLE-Effect + casTLE-Score. (**H**) Candidate hits of the final round of enrichment visualized by a x/y ranked plot using a combined ESP score. (**I**) Trajectories of highest-ranking candidates are plotted over the consecutive rounds of enrichment rounds; size of the bubble represents the pValue (−log10).

The online version of this article includes the following source data and figure supplement(s) for figure 1:

**Source data 1.** Pooled CRISPR activation (CRISPRa) library source list.

**Figure supplement 1.** Evaluation of sunCas9 CRISPR activation (CRISPRa)-mediated transcriptional activation, pooled CRISPRa library quality control, and IL-2 benchmark screens.

---

screens (*Doench, 2018*). Libraries of cells, each with an sgRNA targeting one receptor, can be easily stored and screened for binding to soluble ligands.

Here, we describe a CRISPRa enrichment workflow that employs customized, pooled cell surface receptor sgRNA libraries in combination with magnetic bead-based selection (magnetic-activated cell sorting [MACS]) to enrich for receptor-expressing cells. This approach allows cost-efficient parallel screening with multiple ligands. We created two cell libraries, comprising all single-pass transmembrane (TM1) and multi-pass transmembrane (TM2+) receptors, and screened them with a collection of secreted ligands. To define a set of high-priority ligands, we first curated the human-secreted proteome and selected and expressed 20 at levels sufficient for screening the TM1 and TM2+ libraries. We identified new receptor candidates in more than half of these screens. These were validated using surface plasmon resonance (SPR) and/or cell binding assays. These studies define new receptors for several secreted ligands that function in the immune and nervous systems and provides a resource for future investigations of relationships between the human-secreted and membrane proteome.

## Results
### A CRISPR activating enrichment screening platform

A large variety of different second-generation dCas9 activator (CRISPRa) systems are currently in use. A recent study directly compared a large number of published Cas9 activators systems and found the VPR, SunTag, and SAM approaches perform equally effective across a number of different cell lines and target genes (*Chavez et al., 2016*; *Kampmann, 2018*). The SunTag system was initially developed in K562 cells (myeloid leukemia cell line K562) (*Tanenbaum et al., 2014*), a highly characterized (ENCODE, The Protein Atlas) and easy to handle suspension cell line that is widely used for CRISPR screens and highly suitable for MACS-based applications. CRISPR-mediated activation of transcription using the SunTag system (termed sunCas9) is a precise and scalable method for inducing expression of endogenous genes across a high dynamic range (*Gilbert et al., 2014*). This system uses a dead Cas9 (dCas9) variant fused to a SunTag, a multicopy epitope tag that recruits the VP64 transcriptional activator via binding to a cytoplasmic scFV-nanobody-VP64 fusion protein. sgRNAs guide this complex to the enhancer region of the gene of interest and facilitate target-specific gene activation and expression (*Tanenbaum et al., 2014*).

To evaluate the performance and feasibility of CRISPRa-mediated transcriptional activation of cell surface proteins for a receptor/ligand interaction discovery platform, we first selected 10 well-characterized cell surface receptors with varying mRNA expression levels ranging from not detected to highly expressed in K562 human myeloid leukemia cells (*Figure 1—figure supplement 1A*; *Thul et al., 2017*; *Uhlén et al., 2019*). We then generated a pooled lentiviral mini-library of 10 sgRNAs per enhancer (100 sgRNA elements), matched with 100 control sgRNAs derived from scrambled sequences (*Gilbert et al., 2014*) and transduced K562 cells stably expressing the sunCas9 system

(*Figure 1—figure supplement 1A*). Each library plasmid contained a single sgRNA targeting one of the 10 genes, a GFP fluorescent marker and a puromycin resistance marker. The library-transduced cells were puromycin selected for 5 days to obtain >90% GFP positive cells. The expression levels of the 10 cell surface receptors were then evaluated by cell surface staining (CSS) using APC- (allophyco-cyanin) labeled antibodies against the respective targets (CD122 was used as a control). In summary, the majority (8 out of 10) targets showed elevated cell surface expression to varying degrees in comparison to non-transduced K562 sunCas9 cells or a control receptor (CD122) that was not part of the mini-library (*Figure 1—figure supplement 1A-B*).

We then used human interleukin 2 (IL-2), which has a high affinity receptor subunit termed CD25 (IL2RA), to validate our screening workflow in two parallel screens, simulating two library sizes by diluting the 10 target (100 sgRNA) K562 sunCas9 mini-library by 1:20 and 1:200 with non-transduced K562 sunCas9 cells, corresponding to final library sizes of 200 and 2000 targets, respectively (*Figure 1—figure supplement 1C-F*). Both library pools were incubated with magnetic streptavidin microbeads complexed with biotinylated IL-2, and IL-2 binding cells were isolated in a positive selection workflow by MACS, using Miltenyi LS-MACS columns (*Figure 1—figure supplement 1C-D*). After labeling, washing, and elution, positively selected cells were expanded and stained using an APC-labeled anti-CD25 antibody (*Figure 1—figure supplement 1D*). Genomic DNA was extracted from both consecutive rounds of selection as well as the naïve K562 sunCas9 mini-library itself, followed by barcoding and deep sequencing for both libraries.

Deep sequencing data for each round of selection was analyzed and hits were identified using the robust casTLE statistical framework (*Morgens et al., 2016*) and results were analyzed, filtered, and visualized. We calculated casTLE metrics for each round of selection in comparison to the naïve library. Using casTLE, both IL-2 CRISPRa screens successfully identified IL-2 receptor alpha (IL2RA; CD25) as the top hit with the highest confidence (casTLE Score), casTLE Effects, and significance (pValue) (*Figure 1—figure supplement 1E*). Side by side comparison of enrichment scores for both rounds of selections from both libraries was plotted as bar graphs (*Figure 1—figure supplement 1F*).

## Customized, pooled CRISPRa cell surface receptor library design

Having established the screening workflow (*Figure 1E*, *Figure 1—figure supplement 1*), we sought to leverage the power and efficiency of customized, pooled CRISPRa cell surface libraries to perform targeted screens with secreted orphan ligands. We first compiled a comprehensive list of cell surface receptors by carefully curating the human membrane proteome (*Figure 1A*). We chose a targeted cell surface library approach instead of a genome-wide approach, allowing a smaller library size, resulting in a better signal-to-noise ratio (SNR) and avoiding unwanted transcriptional upregulation of non-membrane proteins. We utilized several databases including HUGO, UniProt, the Human Protein Atlas, and bioinformatic tools (SignalP, TMHMM) to compile two cell surface target lists covering both TM1 and TM2+ cell surface proteins (*Figure 1C*). For the CRISPRa-mediated transcriptional activation of cell surface proteins, we synthesized and cloned two pooled sgRNA libraries, a TM1 and a TM2+ library, each with 10 sgRNAs per target (*Gilbert et al., 2014*). Both libraries include matched controls targeting genomic locations without annotated function (*Figure 1C, D*). K562 cells stably expressing the sunCas9 system were infected with both libraries (TM1; TM2+) at low, medium, and high multiplicity of infection (MOI), then selected with puromycin until the cell population was at least 90% GFP positive, indicating the presence of lentivirus. Cells were recovered and expanded, and representative aliquots were saved as naïve library stocks in liquid nitrogen with at least ×1000 cell number coverage per sgRNA to maintain maximum library complexity. Sufficient sgRNA representation of the naïve library was confirmed by deep sequencing after puromycin selection and showed the highest coverage and diversity at low MOI with at least 91% of reads with at least one reported alignment (R=0.97) for both libraries (*Figure 1—figure supplement 1G, H*). Library information including sgRNA target IDs and sequences for both libraries (TM1; TM2+) can be found in *Figure 1—source data 1*.

## CRISPRa benchmark screen using human IL-2

After library cloning and validation, we benchmarked the sensitivity and robustness of our screening platform with a proof of concept screen using human IL-2, following the protocols used for the mini-library (*Figure 1—figure supplement 1*). We successfully recovered CD25 (IL2RA) as the top hit after two rounds of enrichment, deep sequencing, and analysis following the outlined workflow (*Figure 1E*).

Initially, the naïve TM1 library showed no positive IL-2 binding by tetramer staining (*Figure 1F*). Library enrichment was monitored by IL-2 tetramer staining (IL-2:SA647; 200 nM) throughout the selection workflow, and after only one round of positive selection we observed a significant enrichment of IL-2 selected cells, from 0.1% to 13.3% IL-2 tetramer positive cells (*Figure 1F*; middle FACS plot). After expanding the cells from the first round and subjecting them to a second round of CRISPRa enrichment screening, we observed a further robust increase of IL-2 tetramer positive from 13.3 to 95% IL-2 tetramer positive cells (*Figure 1F*; right FACS plot).

After each consecutive round of selection, enriched cells were expanded and genomic DNA was extracted, followed by barcoding and deep sequencing. Genomic DNA from the K562 sunCas9 TM1 naïve library itself served as the baseline. Following deep sequencing, data from both rounds of consecutive IL-2 selections was analyzed and visualized using the casTLE statistical framework. To predict high-confidence interaction pairs from each dataset, a custom score was then computed for each potential interaction pair by combining all three metrics (casTLE Score, casTLE Effect, and pValue) into one ESP score: (casTLE-Effect + casTLE Score)/pValue. CD25 was identified as the top hit with the highest confidence (casTLE Score), pValue (significance), and casTLE Effect (*Figure 1G, H*). Furthermore, we used the casTLE ESP metrics from each round to plot trajectories of CD25, which allows for a direct evaluation of sgRNA enrichment throughout the selection workflow and shows a positive trajectory for CD25 in the selection workflow (*Figure 1I*), validating the sensitivity and robustness of our screening pipeline.

## Selection and production of secreted proteins for CRISPRa screening

We first generated a secreted proteome master list from several databases, including HUGO, UniProt, the Human Protein Atlas (*Uhlén et al., 2019*), and bioinformatic tools (SignalP, TMHMM) to identify potential high-priority secreted proteins for our screening workflow. After curation of the human-secreted proteome (*Figure 1A, B*), approximately 60% of the ~1600 genes were classified as encoding enzymes (mostly proteases), enzyme inhibitors, serum proteins, or components of saliva, tears, or other fluids (these include carrier proteins), structural, extracellular matrix proteins, antimicrobial proteins, complement factors, coagulation factors, lectins, or unknown. The remaining ~40% of genes were identified as likely to encode secreted ligands acting through cell surface receptors and further examined through literature searches. We classified products of 419 genes as ligands with known receptors that can adequately account for their biology. Finally, we identified 206 gene products either as 'orphans' with no identified receptor or as ligands that are likely to have additional, as yet unidentified receptors in addition to those that have been described. From these 206, we ultimately selected a total of 80 high-priority targets (one per gene; we did not consider isoforms generated through alternative splicing). These had a wide range of tissue expression with many of the chosen secreted ligands being expressed in brain tissue (*Figure 2A*) covering a broad range of molecular function and processes (*Figure 2—figure supplement 1A, B*). Coding sequences for these 80 secreted proteins of interest (SPOI) were synthesized, subcloned into an Avi-6xHIS expression plasmid (*Figure 2—source data 1*), expressed in Expi293F cells, purified with Ni-NTA resin, then biotinylated in vitro and further purified by size-exclusion chromatography (SEC) (*Figure 2—figure supplements 2 and 3*).

## CRISRPa enrichment screens reveal new secreted ligand-receptor interactions

We obtained sufficiently high expression levels for 20 of the 80 high-priority targets (*Figure 2A*, *Figure 2—source data 2*) to allow screening using our cell-based CRISPRa enrichment workflow. In addition to the Expi293F expression system, we explored whether insect expression could serve as viable alternative expression strategy to rescue some of the secreted ligands that failed to express in the Expi293F expression system. To test this strategy, we selected a total of 12 candidates that were subcloned for expression in insect cells (Hi5): 10 candidates that showed no expression from Expi293F cells and an additional 2 candidates that were previously successfully expressed in Expi293F (CC134_HUMAN; SPRC_HUMAN). While the control candidates (CC134_HUMAN; SPRC_HUMAN) showed some low level of expression from insect cells, none of the other 10 candidates tested showed any positive expression (*Figure 2—source data 2*). Names and mRNA expression patterns in normal tissue for these 20 ligands are shown in *Figure 2A*. The 20 high-priority targets show a wide spectrum

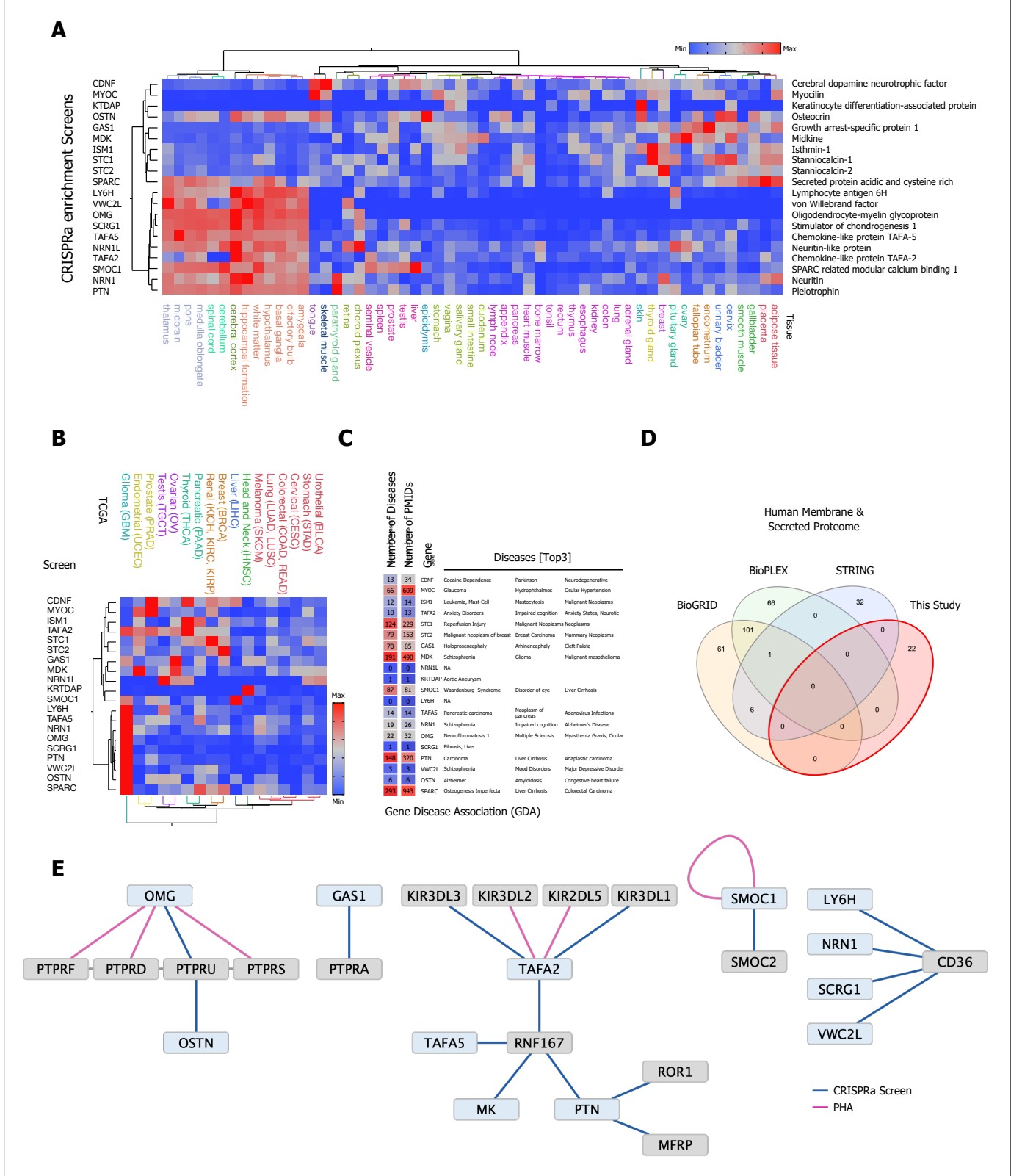

**Figure 2.** Selection and protein production of high-priority secreted ligands and overview of screening results. (**A**) Eighty high-value orphan secreted proteins were selected, synthesized, cloned, and expressed in Expi293F cells, 20 secreted ligands with a broad range of tissue expression passed our quality control and were used in our CRISPR activation (CRISPRa) enrichment workflow. (**B**) TCGA cancer tissue RNA-seq data was obtained for all 20 screened secreted ligands from 17 cancer types representing 21 cancer subtypes and were processed as median FPKM (number fragments per kilobase

*Figure 2 continued on next page*

*Figure 2 continued*

of exon per million reads) and visualized as a hierarchical clustering heatmap. (**C**) Gene disease association (GDA) data for all 20 high-priority candidates used in the CRISPRa enrichment screening workflow: Number of involved Diseases and Publications as well as top three ranking GDA are indicated. Data was obtained from the DisGenet database, full GDA analysis is visualized in *Figure 2—figure supplement 4*, and a fully annotated dataset can be found in *Figure 2—source data 3*. (**D**) Venn diagram visualizing the overlap between physical interactions between secreted and membrane proteins presented in this study and public interaction databases (BioGRID, BioPLEX, and STRING). (**E**) CRISPRa enrichment screening interactions represented as a protein interaction network, nodes represent CRISPRa query secreted ligands (blue) and candidate hits (gray). Edges represent the interactions between nodes. The visualized network shows 22 interactions between secreted and membrane proteins; 18 new interactions from eight screens in the single-pass transmembrane (TM1) and 4 interactions from four screens in the multi-pass transmembrane (TM2+) library between secreted and membrane proteins. Interactions (edges) resulting from CRISPRa enrichment screens are represented in dark blue, interactions resulting from phylogenetic homology analysis (PHA) are visualized in purple.

The online version of this article includes the following source data, source code, and figure supplement(s) for figure 2:

**Source code 1.** Annotated Cytoscape node/edge Gene Disease Association (GDA) network.

**Source data 1.** Plasmid sequences.

**Source data 2.** Secreted protein library.

**Source data 3.** Gene disease association (GDA) data.

**Source data 4.** CRISPR activation (CRISPRa) casTLE statistical analysis.

**Source data 5.** Surface plasmon resonance (SPR) conditions.

**Figure supplement 1.** High-value secreted orphan ligands.

**Figure supplement 2.** Size-exclusion chromatography.

**Figure supplement 3.** Size-exclusion chromatography.

**Figure supplement 4.** Gene disease association (GDA) high-priority secreted ligands.

---

of tissue expression in normal, healthy tissue (*Figure 2A*) and a broad range of expression in cancer (*Figure 2B*) with several candidates enriched in brain tissue and clustering in glioma (GBM, brain tissue, and in various locations in the nervous system, including the brain stem and spinal column). In addition, the majority of targets show a strong GDA with many reported diseases (*Figure 2C*, *Figure 2—figure supplement 4*, *Figure 2—source data 3*).

Each of the 20 secreted ligands was used to screen the TM1 and TM2+ libraries with up to three consecutive rounds of selection, followed by deep sequencing and statistical analysis using the casTLE framework (*Figure 2—source data 4*). In general, screening results of the final round of enrichment were subjected to a first-pass filter using the following cut-offs: casTLE-Effect > 2, casTLE-Score > 2, pValue < 0.05, candidates outside the secreted and membrane proteome and common false positive contaminants were removed. To integrate data analysis and visualization, we used the combined ESP score to rank sort interaction pairs for every screen. In addition, we utilized the aforementioned trajectory plots, which allow for direct evaluation of a potential candidate's enrichment throughout the selection workflow and easy elimination of false positives.

PPIs often occur between phylogenetically related proteins both within and between subfamilies. Hence, screening results were augmented by using phylogenetic homology analysis (PHA), a strategy that has proven to be highly effective to predict additional PPIs between subfamily members (*Wojtowicz et al., 2020*). Furthermore, in an effort to explore potential functional relationships between orphan-secreted ligands and newly discovered interactions, mRNA expression profiles from healthy tissues were obtained from The Human Protein Atlas (*Karlsson et al., 2021*) to perform hierarchical clustering and correlation analysis.

We selected a subset of CRISPRa enrichment screens with high ranking predicted ESP scores for potential interaction pairs for further validation using orthogonal methods, including SPR and CSS. Bona fide PPIs are expected to display distinct association and dissociation kinetics, which can be observed with high sensitivity by SPR, the 'gold' standard to measure biomolecular interactions. CSS was utilized as an alternative orthogonal validation method to show PPIs in a cellular context using a fluorescent-tetramerization-based approach by flow cytometry for high sensitivity detection of putative PPIs on the cell surface. In a first-pass analysis we selected the validated hits of the CRISPRa enrichment screens and performed database searches to calculate overlaps between our screening results and the aggregate of BioGrid, BioPlex, and STRING databases (physical interactions; membrane and secreted proteome). We observed no overlap between any of these databases

**Table 1.** Summary of the new protein-protein interactions (PPIs) tested in this study. Source indicates whether the PPI was discovered in the indicated CRISPR activation (CRISPRa) enrichment screen (Screen) or by phylogenetic homology analysis (PHA). Assay indicates whether interaction was tested by SPR or CSS (SPR conditions are included in *Figure 2—source data 5* ).

| Interaction | Screen | Hit | Library | Source | Assay |
|---|---|---|---|---|---|
| GAS1-PTPRA | GAS1 | PTPRA | TM1 | Screen | SPR, CSS |
| OMG-PTPRD | OMG | PTPRD | TM1 | PHA | SPR |
| OMG-PTPRF | OMG | PTPRF | TM1 | PHA | SPR |
| OMG-PTPRS | OMG | PTPRS | TM1 | PHA | SPR |
| OMG-PTPRU | OMG | PTPRU | TM1 | Screen | SPR |
| OSTN-PTPRU | OSTN | PTPRU | TM1 | Screen | SPR |
| MK-RNF167 | MK | RNF167 | TM1 | Screen | SPR |
| PTN-RNF167 | PTN | RNF167 | TM1 | Screen | SPR |
| PTN-ROR1 | PTN | ROR1 | TM1 | Screen | SPR |
| PTN-MFRP | PTN | MFRP | TM1 | Screen | SPR |
| SMOC1-SMOC1 | SMOC1 | SMOC1 | TM1 | PHA | SPR |
| SMOC1-SMOC2 | SMOC1 | SMOC2 | TM1 | Screen | SPR |
| TAFA2-KIR3DL1 | TAFA2 | KIR3DL1 | TM1 | Screen | SPR, CSS |
| TAFA2-KIR3DL2 | TAFA2 | KIR3DL2 | TM1 | PHA | SPR, CSS |
| TAFA2-KIR3DL3 | TAFA2 | KIR3DL3 | TM1 | Screen | CSS |
| TAFA2-KIR2DL5A | TAFA2 | KIR2DL5A | TM1 | PHA | CSS |
| TAFA2-RNF167 | TAFA2 | RNF167 | TM1 | Screen | SPR |
| TAFA5-RNF167 | TAFA5 | RNF167 | TM1 | Screen | SPR |
| LY6H-CD36 | LY6H | CD36 | TM2+ | Screen | CSS |
| NRN1-CD36 | NRN1 | CD36 | TM2+ | Screen | CSS |
| SCRG1-CD36 | SCRG1 | CD36 | TM2+ | Screen | CSS |
| VWC2L-CD36 | VWC2L | CD36 | TM2+ | Screen | CSS |

PHA = phylogenetic homology analysis. SPR = surface plasmon resonance. CSS = cell surface staining.

and the hits reported in this study (*Figure 2D*). As we previously reported for interactome screens of *Drosophila* and human cell surface proteins (*Özkan et al., 2013*; *Wojtowicz et al., 2020*), high-throughput PPI analysis methods such as Y2H and AP/MS generate mostly false positive interactions for secreted and membrane proteins and are unable to identify genuine interactions found through ELISA and/or cell-based screening methods.

In total, we tested 22 candidate PPIs between the secreted and membrane proteome by SPR and/or CSS from 12 screens with PPIs in both the TM1 and the TM2+ library. A cytoscape interaction network of novel PPIs observed in screen is visualized in *Figure 2E* and summarized in *Table 1*, where nodes represent the secreted ligands (blue) and cell surface receptors discovered (gray) and edges represent the interactions between them. These validation data are shown for selected PPIs in *Figures 3–7*.

## Oligodendrocyte-myelin glycoprotein binds to multiple receptor tyrosine phosphatases

Protein tyrosine phosphorylation is a fundamental regulatory step in intracellular signal transduction and is orchestrated in a coordinated fashion by activities of protein tyrosine kinases and protein tyrosine phosphatases (PTPs). PTPs play essential roles in the regulation of growth, differentiation,

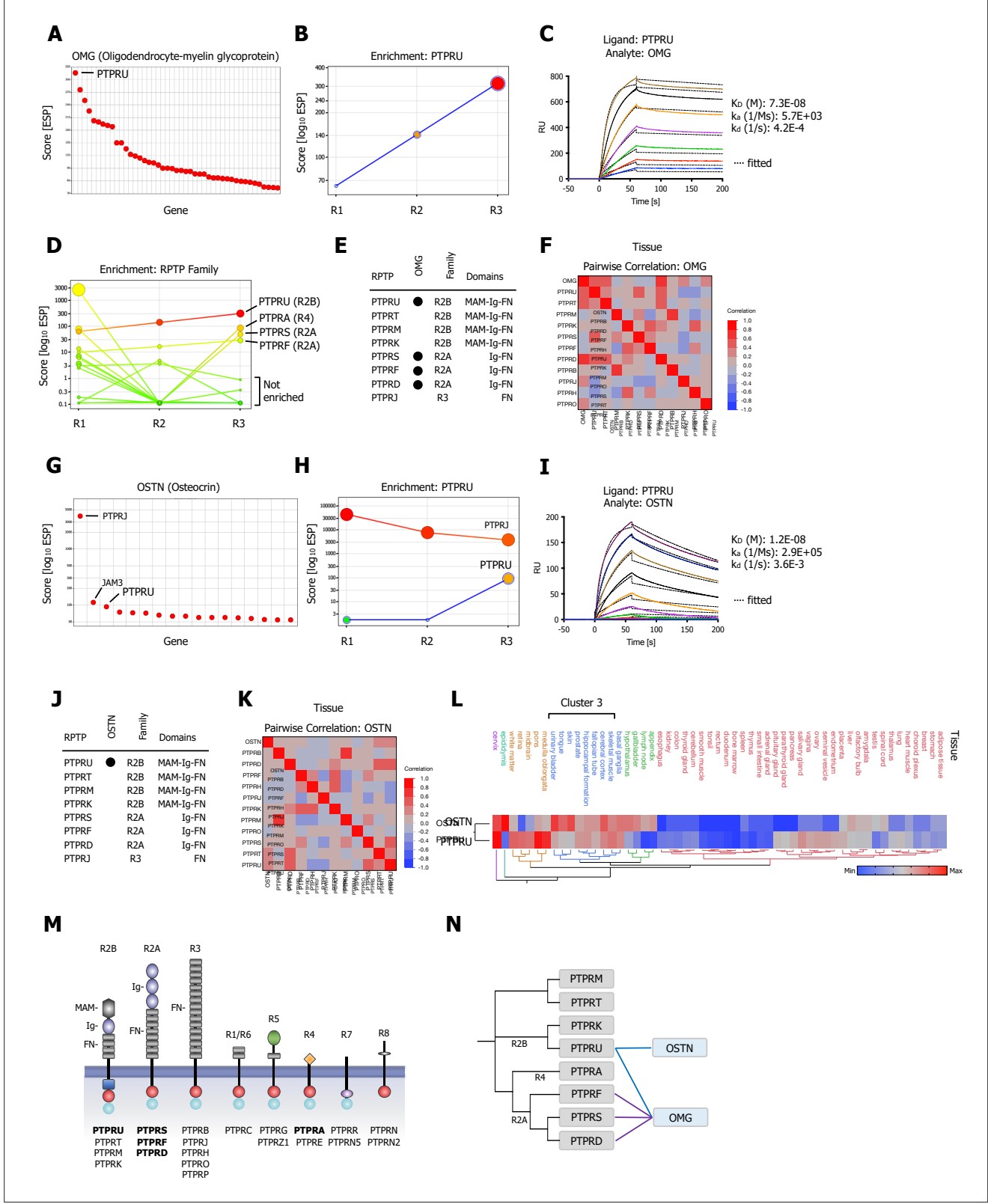

**Figure 3.** CRISPR activation (CRISPRa) screening results and surface plasmon resonance (SPR) validation of Type R2A and Type R2B PTPRs for OMG (Oligodendrocyte-myelin glycoprotein) and OSTN (Osteocrin). (**A**) Ranked x/y scatter plot of Round 3 of the OMG screen. (**B**) Depicts the trajectory of the highest ranking candidate, PTPRU, plotting ESP scores for all three consecutive rounds of selections in a x/y enrichment plot. The size of the bubble represents the pValue (−log10). (**C**) SPR sensorgram and binding kinetics for human PTPRU-ECD (ligand) binding to human OMG (analyte), sensograms

*Figure 3 continued on next page*

*Figure 3 continued*

were fitted using a 1:1 binding model; raw data are shown in color, fitted data are shown as black dotted lines. (**D**) Shows enrichment scores for additional members of the receptor protein tyrosine phosphatase (RPTP) family members found in the OMG screen. (**E**) Summary of SPR results testing binding of R2A, R2B, and R3 RPTP subfamily members (*Figure 3—figure supplement 1A*). (**F**) Multivariate heatmaps for OMG calculated from normal tissue mRNA expression correlations. (**G**) OSTN screen ranked x/y scatter plot of Round 3 ESP scores; top three candidates as indicated. (**H**) Trajectory of high ranking candidates PTPRJ and PTPRU by plotting ESP scores for all three consecutive rounds of selections in an x/y enrichment plot; size of the bubble represents the pValue (−log10). (**I**) SPR sensorgram and binding kinetics for human PTPRU-ECD (ligand) binding to human OSTN (analyte), sensograms were fitted using a 1:1 binding model; raw data are shown in color, fitted data are shown as black dotted lines. (**J**) Summary of SPR results (*Figure 3—figure supplement 1C*) testing binding of OSTN (analyte) binding to R2A, R2B, and R3 RPTP subfamily members (ligands). (**K**) Multivariate heatmap for OSTN calculated from normal tissue mRNA expression correlations. (**L**) Hierarchical two-way clustering heatmap of normal tissue mRNA expression for OSTN and PTPRU. (**M**) Schematic representation of the domain architectures of RPTP subfamilies; PPIs as indicated. (**N**) Dendrogram of PTPR R2A, R2B, R4 subfamily members calculated from multiple sequence alignments (MSA) and visualization of the reported interactions in a node/edge network format, secreted ligands (blue) and cell surface receptor protein-protein interactions (PPIs) observed (gray). Interactions (edges) resulting from CRISPRa enrichment screens are represented in dark blue, interactions resulting from phylogenetic homology analysis (PHA) are visualized in purple.

The online version of this article includes the following figure supplement(s) for figure 3:

**Figure supplement 1.** PTPR subfamily-related surface plasmon resonance (SPR) sensorgrams for oligodendrocyte-myelin glycoprotein (OMG) and OSTN and hierarchical clustering analysis of normal tissue for OMG.

oncogenic transformation, and other processes (*Julien et al., 2010*). The classical PTPs include cytoplasmic PTPs and transmembrane RPTPs, which can be classified into distinct subfamilies according to their domain architecture. Most RPTPs display features of cell adhesion molecules with a diverse domain repertoire including MAM (meprin, A-5) domains, Ig (immunoglobulin-like) domains, and FN (fibronectin) Type III repeats in their extracellular segments (*Figure 3M*; *Tonks, 2006*). In our human in vitro interactome screen, we identified new cell surface binding partners for multiple RPTPs that are likely to mediate cell-cell and/or cell-matrix interactions (*Wojtowicz et al., 2020*).

In our CRISPRa screen with oligodendrocyte-myelin glycoprotein (OMG), we observed the Type R2B subfamily member PTPRU as the top-ranking hit (*Figure 3A*) with a positive enrichment trajectory over all three rounds of selection (*Figure 3B*). We confirmed binding of OMG to PTPRU by SPR, with a $K_D$ of ~70 nM (*Figure 3C*). We also identified two members of the R2A subfamily, PTPRF and PTPRS, as well as the R4 subfamily member PTPRA as enriched in the OMG screen (*Figure 3D*). Type R2A (PTPRD, PTPRF, PTPRS), R2B (PTPRK, PTPRM, PTPRT, PTPRU), and R3 (PTPRB, PTPRH, PTPRJ, PTPRO, PTPRP) are the largest RPTP subfamilies. They all share large ECDs that include FN-III repeats. R2A RPTPs also have Ig domains, and R2B RPTPs have both Ig and MAM domains (*Figure 3M*). PPIs often occur between phylogenetically related proteins both within and between subfamilies. We examined binding of OMG to all R2A and R2B subfamily members as well as PRPRJ (R3) by SPR. Binding in the micromolar affinity range was observed for all three R2A RPTPs (PTPRD, PTPRF, PTPRS) but only for PTPRU among R2B RPTPs (*Figure 3E*; *Figure 3—figure supplement 1A*). Hierarchical clustering by healthy tissue expression correlations may infer functionally related communities. We therefore examined healthy tissue mRNA expression profiles for OMG, R2A, R2B, and R3 RPTP family members from the Human Protein Atlas (*Karlsson et al., 2021*) and performed a multivariate clustering analysis. OMG clustered with several RPTP family members including binding partners PTPRU and PTPRD (*Figure 3F*). In the nervous system, these RPTPs are expressed primarily in neurons, and could function as receptors for OMG, which is expressed in oligodendrocytes and some neurons (*Figure 3—figure supplement 1B*).

## PTPRU binds to Osteocrin (OSTN), a primate-specific brain ligand

PTPRU was also identified as a potential hit in a screen for Osteocrin (OSTN). Although our initial ESP ranking showed PTPRJ, a RPTP member of the R3 subfamily, as the top-ranking hit for OSTN (*Figure 3G*), analyzing the enrichment trajectories over all three rounds of selections revealed that PTPRJ actually followed a negative trajectory (*Figure 3H*). By contrast, the R2B family member PTPRU showed a significant positive enrichment trajectory over the course of the screening workflow compared to PTPRJ (*Figure 3H*). We therefore analyzed binding of OSTN to a panel of R2A, R2B, and R3 RPTP members and found that OSTN exclusively bound to PTPRU, with a $K_D$ of ~12 nM (*Figure 3I, J*) and none of the other R2A or R2B subfamily members (*Figure 3—figure supplement 1C*).

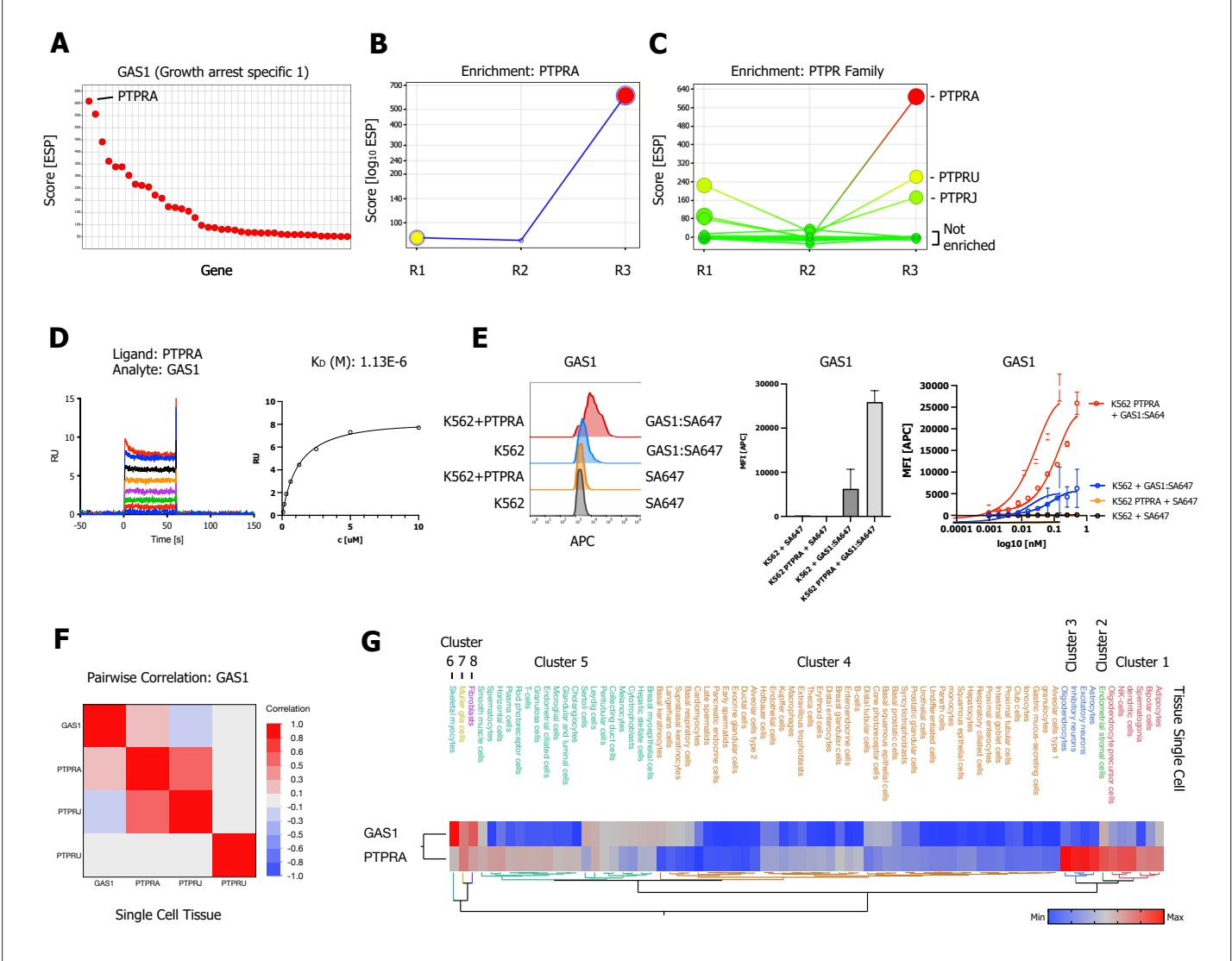

**Figure 4.** Growth arrest specific 1 (GAS1) CRISPR activation (CRISPRa) enrichment screen identifies receptor protein tyrosine phosphatase (RPTP) subfamily member PTPRA. (**A**) Ranked x/y scatter plot for the GAS1 CRISPRa enrichment screen (ESP scores). (**B**) Trajectory plot of the highest ranking candidate PTPRA for all three consecutive rounds of selections in an x/y enrichment plot, size of the bubble represents the pValue (−log10). (**C**) Comparison of ESP trajectories for PTPRA and two lower scoring RPTP subfamily members (PTPRU, PTPRJ). (**D**) Surface plasmon resonance (SPR) sensorgram and steady-state curve for human GAS1 (analyte) binding to PTPRA-ECD (ligand) binding in comparison to PTPRU and PTPRJ (*Figure 4— figure supplement 1A, B*; no binding observed). (**E**, *Figure 4—figure supplement 1C, B*) Cell surface staining of K562 (untransduced) or K562 cells lentivirally transduced with FLAG-tagged full-length PTPRA with GAS1:SA-647 tetramers (400 nM) and analysis by flow cytometry, representative FACS histograms, quantification, and full titration (1:1 dilutions; 400 nM tetramer). Data are represented as mean ± SD (n=3). (**F**) Multivariate heatmaps for GAS1 and the PTPRA, PTPRU, and PTPRJ calculated from single-cell normal tissue mRNA expression correlations. (**G**) Hierarchical two-way clustering heatmap of single-cell normal tissue mRNA expression for GAS1 and PTPRA.

The online version of this article includes the following figure supplement(s) for figure 4:

**Figure supplement 1.** Growth arrest specific 1 (GAS1) screen-related surface plasmon resonance (SPR) sensorgrams.

OSTN (Musclin) is a 130 aa peptide hormone that was originally identified in mouse bone and muscle. It regulates bone growth, supports physical endurance, and mediates diverse cardiac benefits of physical activity (*Subbotina et al., 2015*). These actions could be mediated through OSTN's binding to the natriuretic peptide clearance receptor (NPR-C) (*Moffatt et al., 2007*). By binding to NPR-C, OSTN decreases clearance of natriuretic peptides and thereby increases signaling through the NPR-A and NPR-B receptors. In primates, however, the OSTN gene has acquired neuron-specific

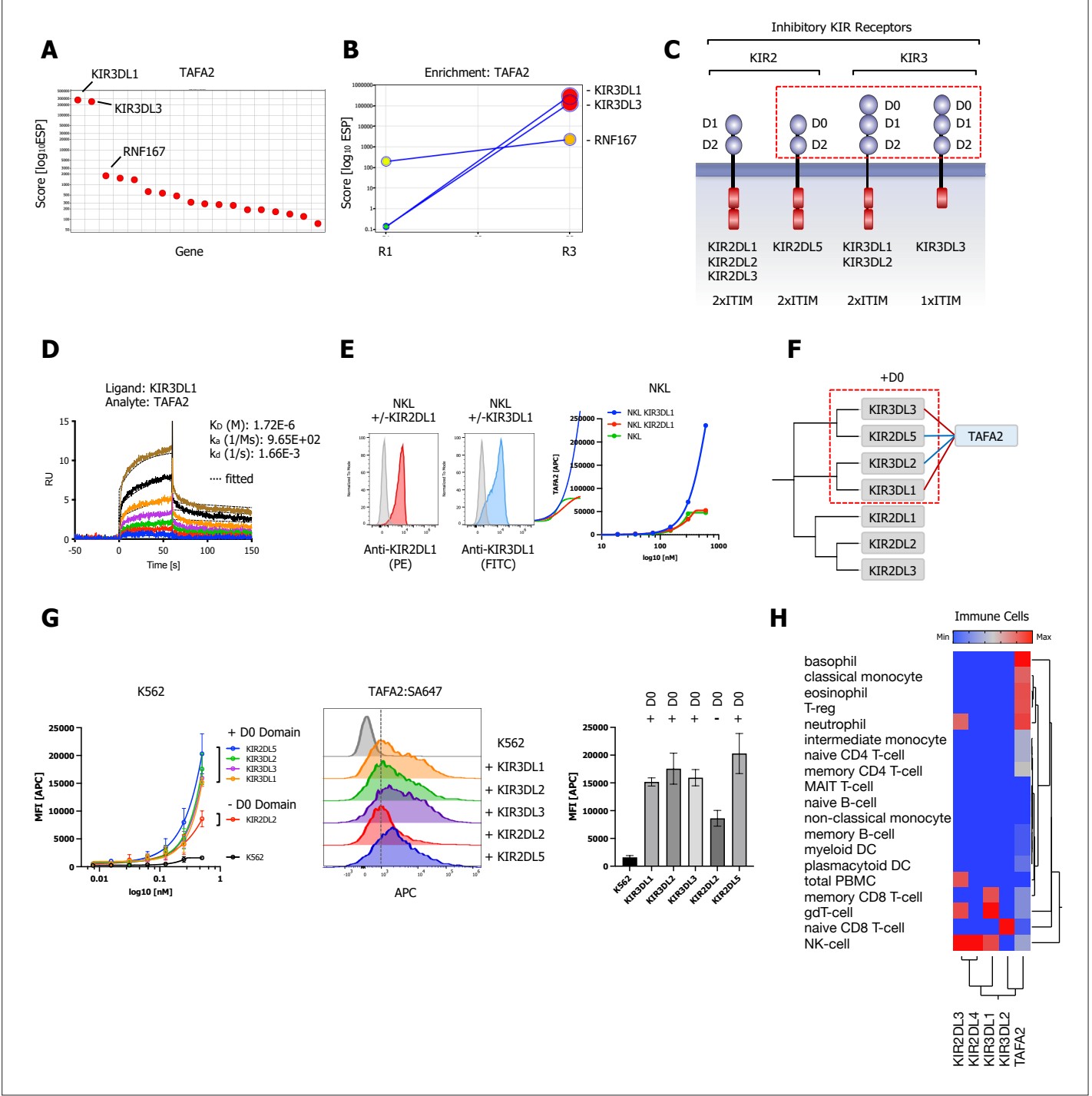

**Figure 5.** Killer immunoglobulin-like receptor (KIR) subfamily protein-protein interactions (PPIs) identified by CRISPR activation (CRISPRa) screening and phylogenetic homology analysis (PHA) approach for TAFA2. (**A**) ESP rank plot of the TAFA2 CRISPRa enrichment screen. (**B**) Trajectory plot of the top three ranking candidates for all consecutive rounds of selections in a x/y enrichment plot, size of the bubble represents the pValue (−log10). (**C**) Schematic representation of the domain architecture of KIR2 and KIR3 subfamily members of inhibitory KIRs. (**D**) Surface plasmon resonance (SPR) sensorgrams and binding kinetics for human TAFA2 (analyte) binding to KIR3DL1-ECD (ligand), sensorgrams were fitted using a 1:1 binding model; raw data are shown in color, fitted data are shown as black dotted lines. (**E**) Cell surface staining of NKL or NKL cells expressing KIR3DL1, KIR2DL1 with TAFA2:SA-647 tetramers (200 nM) and analysis by flow cytometry. (**F**) Dendrogram of the KIR2 and KIR3 subfamily calculated from multiple sequence alignments of KIR ECDs (gray) and PPIs (TAFA2; blue) observed in the CRISPRa screen (red) and predicted by PHA (blue). (**G**) Cell surface staining of K562 control cells or K562 cells lentivirally transduced with full-length KIR3DL1, KIR3DL2, KIR3DL3, KIR2DL2, or KIR2DL5A (FLAG-tagged; *Figure 5—figure supplement 1C*) with TAFA2:SA-647 tetramers (200 nM) and analysis by flow cytometry: full titration (1:1 dilutions; 200 nM tetramer),

*Figure 5 continued on next page*

Figure 5 continued
representative FACS histograms and quantification (200 nM tetramer). Data are represented as mean ± SD (n=3). (**H**) Hierarchical two-way clustering heatmap of immune cell mRNA expression data for TAFA2, KIR2, and KIR3 subfamily members.

The online version of this article includes the following figure supplement(s) for figure 5:

**Figure supplement 1.** TAFA2 screen-related surface plasmon resonance (SPR) sensograms for KIR3 family members.

regulatory elements, and primate OSTN is expressed in cortical neurons and is induced by depolarization in in vitro cultures and by sensory stimuli in vivo. OSTN restricts dendritic growth after depolarization. OSTN expression peaks during the onset of synaptogenesis in fetal development, but it continues to be expressed in neocortex in adults (*Ataman et al., 2016*). A pairwise correlation of normal human tissue mRNA expression data for OSTN and the R2A, R2B, and R3 RPTP subfamilies showed correlation only with PTPRU and its close relative PTPRT (*Figure 3K*). A hierarchical cluster analysis of human tissue mRNA expression data shows a strong correlation of OSTN and PTPRU in brain and skeletal muscle (*Figure 3L*; Cluster 3).

A PHA of the R2A, R2B, and R4 PTPR subfamily was combined with a summary of PPIs identified for OMG and OSTN as a node/edge network (*Figure 3N*). While the R2A subfamily members cluster closely within their family, the R2B subfamily appears to be more diverse with two subclusters (PTPRM and PTPRT, PTPRK, and PTPRU) and might be due to the fact that MAM domains are less conserved than FN or Ig domains (*Figure 3—figure supplement 1D*).

## The growth arrest specific 1 protein (GAS1) binds to PTPRA

PTPRA was identified as the highest-ranking hit in the growth arrest specific 1 (GAS1) screen, with the highest ESP score and a consistent positive enrichment over three rounds of positive selection (*Figure 4A, B*), followed by lower scoring PTPRU and PTPRJ with low enrichment trajectory (*Figure 4C*). PTPRA is a member of the R4 RPTP subfamily (see also *Figure 3M*), which have short, highly glycosylated ECDs (*Tonks, 2006*). GAS1 bound exclusively to PTPRA, with a $K_D$ of ~1 μM (*Figure 4D*). No binding was observed to PTPRU or PTPRJ (*Figure 4—figure supplement 1A, B*) by SPR. We also showed that tetramerized GAS1 (GAS1:SA647) exhibits increased binding to K562 cells that overexpress PTPRA, demonstrating that GAS1 is a soluble ligand for cell surface PTPRA (*Figure 4E*, *Figure 4—figure supplement 1C*). A multivariate clustering showed that GAS1 is clustering more closely to PTPRA than to PTPRU or PTPRJ (*Figure 4F*).

PTPRA is ubiquitously expressed, while GAS1 has a more restricted expression pattern with a stronger correlation with PTPRA in fibroblasts, Muller glia cells, and skeletal myocytes (*Figure 4G*, Clusters 6–8). GAS1 and PTPRA are both involved in RET tyrosine kinase signaling through SRC, as well as in other signaling pathways (*Biau et al., 2013*; *Mustelin and Hunter, 2002*; *Yao et al., 2017*). GAS1 is related to the GFR1 family of transmembrane proteins, which are coreceptors for the RET receptor tyrosine kinase (RTK). RET-GFR1 complexes bind to glial-derived neurotrophic factor (GDNF), leading to RET autophosphorylation and activation of downstream Akt and MAPK signaling pathways. GAS1 interacts directly with RET and recruits it to lipid rafts. GAS1 binding causes a reduction in GDNF-induced Akt phosphorylation, suggesting that it is a negative regulator of RET signaling (*Cabrera et al., 2006*; *López-Ramírez et al., 2008*). PTPRA also associates with RET signaling complexes and can directly dephosphorylate RET, causing inhibition of RET signaling (*Yadav et al., 2020*). Elevated levels of PTPRA leads to promotion of lung cancer and is associated with poor prognosis and overall survival (*Gu et al., 2017*; *Lin et al., 2020*) through c-Src activation. PTPRA has not been demonstrated to directly bind to RET, however, and a linkage between PTPRA and RET might be provided by GAS1.

## TAFA-2 selectively interacts with inhibitory KIRs

A CRISPRa enrichment screen with TAFA-2 (FAM19A2; chemokine-like family member 2) identified two inhibitory killer immunoglobulin-like receptors (KIRs), KIR3DL1 and KIR3DL3, which are selectively expressed on natural killer (NK) cells, as the highest-ranking hits by ESP scoring (*Figure 5A*) with a positive trajectory during the selection workflow (*Figure 5B*). KIRs are a polymorphic subfamily of MHC class I receptors (*Li and Mariuzza, 2014*; *Pende et al., 2019*; *Sivori et al., 2019*). KIR3s have D0, D1, and D2 Ig-like domains. By contrast, KIR2s have only D1 and D2 domains, except for KIR2DL5, which has a D0 and D2 but lacks D1 (*Figure 5C*). The structure of KIR3DL1 complexed to an HLA-B

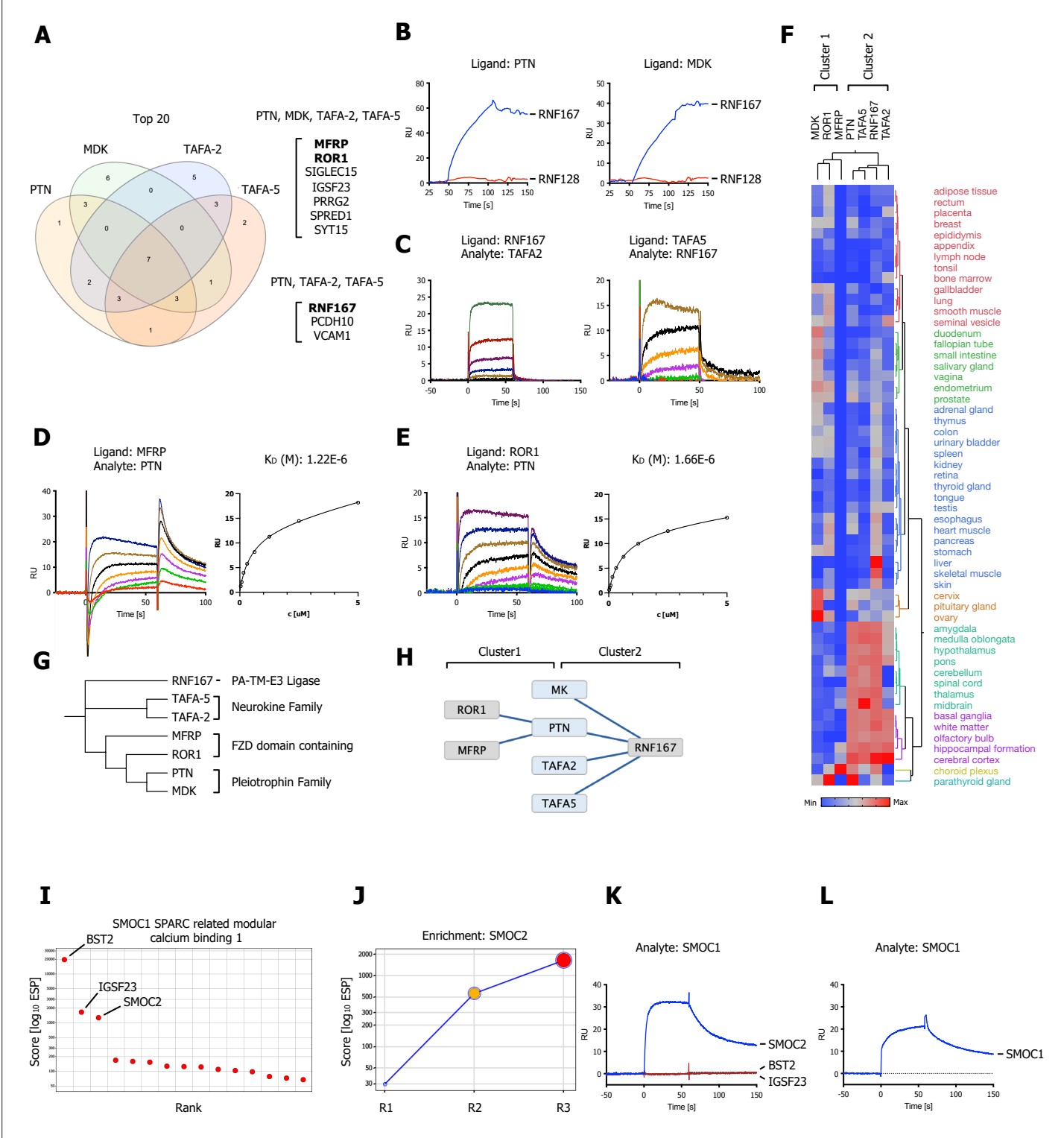

**Figure 6.** Shared hits for the pleiotrophin family (MDL, PTN), neurokine family members (TAFA-2 and TAFA-5), and a SPARC-related ligand (SMOC1). (**A**) Venn diagram depicting overlap of the top 20 ranking candidates for PTN, MK (MDK), TAFA-2, and TAFA-5 screens. Shared hits for PTN, MK, TAFA-2, TAFA-5, and shared candidates for PTN, TAFA-2, and TAFA-5 screens are listed. (**B**) PTN or MDK ectodomains were captured on sensors (ligands) and analyzed for binding to RNF167 (RNF128 was used as a negative control and showed no binding). (**C**) Surface plasmon resonance (SPR) sensorgrams for human TAFA-2 and TAFA-5 binding to RNF167-ECD in comparison to MCAM (*Figure 6—figure supplement 1A*; no binding observed). (**D, E**) SPR sensorgrams and steady-state curves for human PTN (analyte) binding to MFRP-ECD and ROR1-ECD (ligand) in comparison to human MDK (*Figure 6—*

*Figure 6 continued on next page*

*Figure 6 continued*

**figure supplement 1B, C**; no binding observed). (**F**) Hierarchical two-way clustering of mRNA expression data from normal tissue for PTN, MDK, TAFA-2, TAFA-5, RNF167, MFRP, and ROR1. Cluster 1 (MDK, ROR1, MFRP) and Cluster 2 (PTN, TAFA2, TAFA5, RNF167) are indicated. (**G**) Dendrogram of PTN, MDK, TAFA-2, TAFA-5, RNF167, MFRP, and ROR1 calculated from multiple sequence alignments (MSA). (**H**) Visualization of the reported interactions in a node/edge network format with indicated clusters, of secreted ligands (blue) and cell surface receptor protein-protein interactions (PPIs) observed (gray). (**I**) SMOC1 screening results visualized in ranked x/y scatter plot (ESP scores). (**J**) Trajectory plot of SMOC2 for all three consecutive rounds of selections in a x/y enrichment plot, size of the bubble represents the pValue (−log10). (**K, L**) BST2, IGSF23, and SMOC2 ECDs were captured on sensors (ligands) and analyzed for binding to SMOC1 (analyte), SPR assay showing binding of SMOC1 (analyte) to SMOC2 or SMOC1, immobilized on a sensor chip (ligand).

The online version of this article includes the following figure supplement(s) for figure 6:

**Figure supplement 1.** RNF167 and MDK-related surface plasmon resonance (SPR) sensograms.

reveals that the helices and bound peptide of the HLA engage with the D1 and D2 domains of the KIR, while the D0 domain extends down toward the β-2-microglobulin subunit and engages sequences that are highly conserved among all HLA-A and B alleles (*Li and Mariuzza, 2014*).

We observed binding of TAFA-2 to KIR3DL1 to by SPR, with a $K_D$ of ~1.7 µM (*Figure 5D*, *Figure 5—figure supplement 1A*). Binding to KIR3DL3 was at the limit of detection and did not saturate (*Figure 5—figure supplement 1B*). To examine binding at the cell surface, fluorescent TAFA-2 tetramers (TAFA2:SA647) were incubated with NKL or NKL cells expressing either full-length KIR3DL1 or KIR2DL1 (*Figure 5E*; left panels). Flow cytometry analysis revealed concentration-dependent binding of TAFA-2 to cells expressing KIR3DL1, but not to those expressing KIR2DL1 (*Figure 5E*; right panel).

Interestingly, a PHA of inhibitory KIR family members revealed that D0 domain containing KIRs are more closely related by sequence than KIRs without a D0 domain (*Figure 5F*). Hence, to further define KIR binding specificity, we tested binding of TAFA-2 tetramers (TAFA2:SA647) to K562 cells expressing KIR3DL1, KIR3DL2, KIR3DL3, KIR2DL2, or KIR2DL5A. We observed higher concentration-dependent binding of TAFA-2 to cells expressing the KIR3s or KIR2DL5A, which all have D0 domains, compared to KIR2DL2, which does not have a D0 domain (*Figure 5G*, *Figure 5—figure supplement 1C*). These data suggest that D0 domains are required for optimal TAFA-2 binding.

TAFA-2 is a member of a highly conserved 5-gene family (TAFA1-5) of chemokine-like peptides (neurokines) expressed in the brain. Like other chemokines, TAFAs 1, 4, and 5 bind to G protein-coupled receptors. TAFAs 1–4 all complex with neurexins during their passage through the ER/Golgi pathway, leading to formation of disulfide-bonded cell surface neurexin-TAFA complexes (*Khalaj et al., 2020*; *Sarver et al., 2021*; *Tom Tang et al., 2004*). Although TAFA-2 has only been examined in the brain, the gene is also expressed in the immune system. Its expression is restricted to naïve and memory regulatory T cells (T-regs), basophils, and neutrophils, with the highest expression levels being observed in basophils (*Figure 5H*). Neurexins are not expressed in these cell types, so TAFA-2 may be secreted as a monomer or complexed to another protein. The observed interactions of TAFA-2 with D0 domains of KIRs suggest that expression of the chemokine by T-regs or basophils might modulate KIR signaling in NK cells in response to binding of HLA on target cells. In addition to HLA, KIR3DL3 was more recently discovered as a novel interaction partner for HHLA2 (a immune checkpoint member of the B7 family), which has both immune inhibitory and activating abilities and is expressed in many human cancers (*Wei et al., 2021*). Interestingly, KIR3DL3 and TMIGD2, another HHLA2 interaction partner, were simultaneously able to bind to different sites of HHLA2. This would open another intriguing way of regulating signaling KIR3DL3/TMIGD2-HHLA2 with regard to the newly discovered KIR3DL3-TAFA2 interaction.

## Screening results for pleiotrophin family members (MDK, PTN), neurokine superfamily members (TAFA-2, TAFA-5), and a SPARC-related ligand (SMOC1)

In *Figure 6*, we show results of additional screens with shared hits for the pleiotrophin family members midkine (MDK) and pleiotrophin (PTN), and the neurokine family members TAFA-2 and TAFA-5.

We analyzed the top 20 highest ranking candidates for MDK, PTN, TAFA-2, and TAFA5 and found a strong overlap of hits for several candidates (*Figure 6A*). We identified the transmembrane PA-TM-RING E3 ligase RNF167 as shared hits between screens for PTN, TAFA-2, and TAFA-5. PTN and MDK

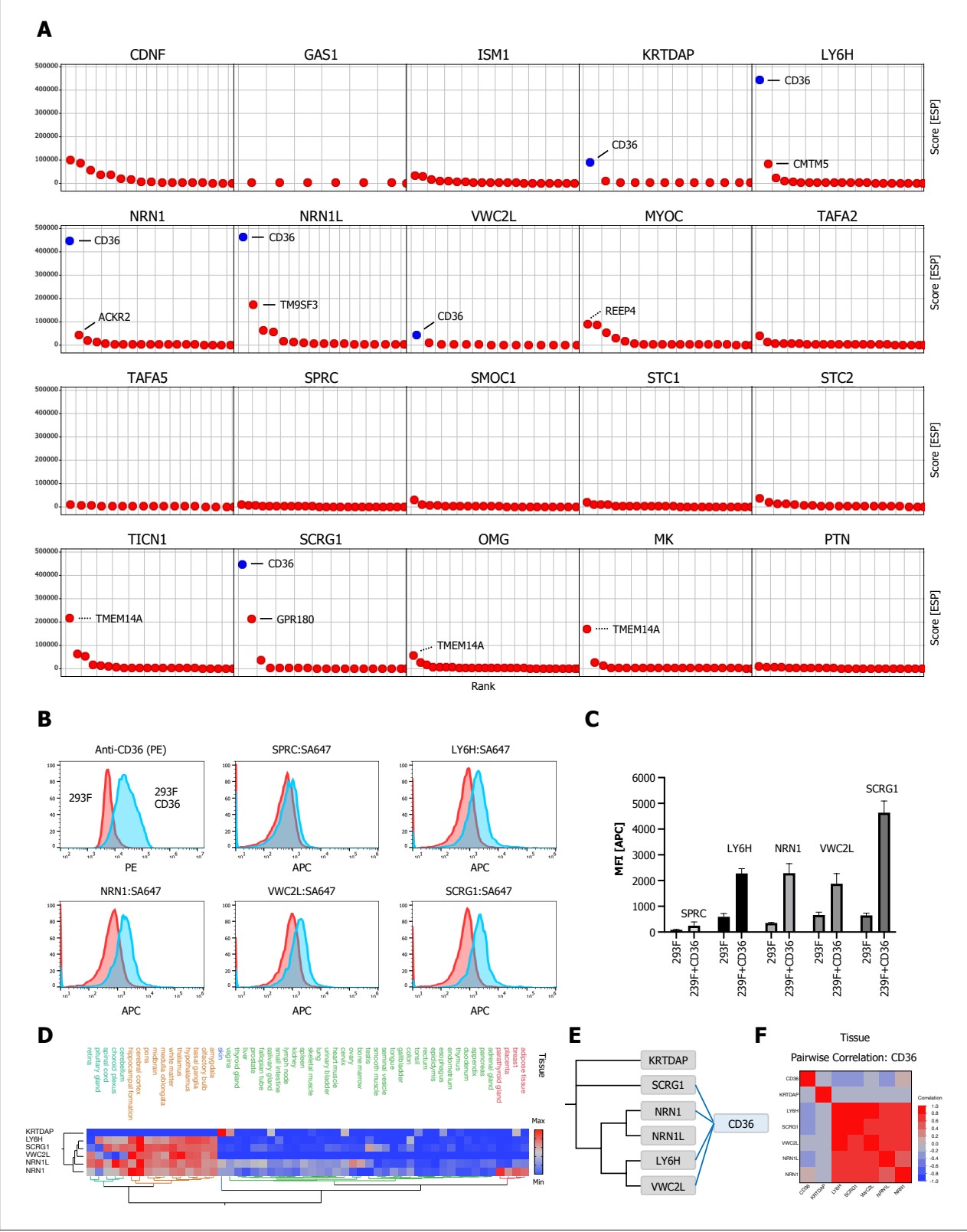

**Figure 7.** Multifunctional scavenger receptor CD36 binds multiple secreted ligands. (**A**) Ranked ESP scatter trellis plots of all analyzed multi-pass transmembrane (TM2+) library screens shows CD36 as the highest ranking hit for multiple screens (CD36 indicated in blue), top two ranking hit candidates are labeled as indicated. False positive candidates are indicated by dotted lines (TMTM14A, REEP4). (**B–C**) Cell surface staining of full-length CD36-transfected and control cells with SA-647 tetramerized (400 nM) LY6H, NRN1, VWC2L, or SCRG1 and analysis by flow cytometry, SPRC

*Figure 7 continued on next page*

*Figure 7 continued*

tetramers (SPRC:SA647) served as a negative (non-CD36 enriched) control. (**B**) Representative histograms (red, control cells; blue, CD36 positive cells). (**C**) Quantification of flow data (Data are represented as mean ± SD ; n=3); data are represented as mean ± SD. (**D**) Hierarchical two-way clustering heatmap of normal tissue cell mRNA expression data for CD36 and the indicated CD36 enriched screens. (**E**) Dendrogram of observed CD36 (blue) protein-protein interactions (PPIs) in the indicated CRISPRa screens (gray) calculated from multiple sequence alignments of the indicated secreted ligands (gray). (**F**) Multivariate correlation analysis of normal tissue cell mRNA expression data for CD36 and the indicated CD36 enriched screens.

The online version of this article includes the following figure supplement(s) for figure 7:

**Figure supplement 1.** Gating strategy of 293F cells.

are two members of the neurite growth-promoting factor family. They are neuromodulators with multiple neuronal functions during development (*González-Castillo et al., 2015*). MDK and PTN have been found to bind to several different cell surface receptors, including RPTPZ, which has a CS-linked ECD, Syndecans (which are linked to HS), a transmembrane low-density lipoproteins (LRP), the RTK ALK, and integrins (*Papadimitriou et al., 2016*).

Like TAFA-2, TAFA-5 is a member of the TAFA superfamily, a emerging family of neurokines that are highly expressed in brain tissue, especially hippocampus, cerebral cortex, white matter, and ganglia (*Figure 6F*). *TAFA5* knockout mice display increased depressive-like behaviors and impaired hippocampus-dependent spatial memory (*Huang et al., 2021*).

We observed binding of RNF167 to PTN and MDK as well as TAFA-2 and TAFA-5 by SPR (*Figure 6B, C*, blue sensogram). No binding was observed to RNF128 which is a related transmembrane E3 ligase of the same subfamily (*Figure 6B, C*, red sensogram) and no binding was observed to RNF167 testing an unrelated receptor ECD (*Figure 6—figure supplement 1A*). RNF167 is a member of the transmembrane PA-TM-RING E3 ligase family with approximately 10 members and exert widespread involvement in several diseases (*Cai et al., 2022*). These E3 ligases are minimally defined by three conserved domains, a protease-associated (PA) domain that acts as a substrate recruitment domain, a transmembrane domain (TM), and a RING-H2 finger (RNF) (*Nakamura, 2011*). In general, the E3 ligase family (~600 predicted RING E3's) still remains highly elusive and the majority of PA-TM-RING E3 ligases remain orphans, mostly due to the nature of the ubiquitylation cascade, which are being characterized by very weak affinity and fast kinetics (*Duan and Pagano, 2021*; *Komander and Rape, 2012*; *Metzger et al., 2014*). RNF167 (also known as Godzilla) is widely expressed in all tissues with enhanced expression in brain tissue (*Figure 6F*) and has recently been implicated in the regulation of the AMPA receptor (AMPAR) (*Ghilarducci et al., 2021*; *Lussier et al., 2012*). A two-way hierarchical clustering of normal tissue mRNA expression data shows a strong correlation between RNF167, TAFA-2, TAFA-2, and PTN in brain tissue (*Figure 6F*; Cluster 2).

The transmembrane RTK ROR1 and the membrane Frizzled (FZD)-related protein (MFRP) were identified as shared candidate hits among several screens: PTN, MDK, TAFA-2, and TAFA-5. We observed binding of PTN to ROR1 and MFRP (*Figure 6D, E*) with $K_D$s between ~1 and 2 µM, no binding was observed for MDK (*Figure 6—figure supplement 1B, C*). Interestingly, both receptors share an evolutionary conserved FZD domain in the ECD (*Yan et al., 2014*). ROR1 and its closely related paralog ROR2 are receptors for Wnt5a and other Wnts in the planar cell polarity pathway (*Endo et al., 2022*; *Green et al., 2014*; *Minami et al., 2010*). MFRP is Type II transmembrane protein with an extracellular FZD domain. MFRP is predominantly expressed in the retinal pigment epithelium with high expression in brain tissue (choroid plexus and required for both prenatal ocular growth and postnatal emmetropization *Katoh, 2001*; *Sundin et al., 2008*). A PHA of the involved PPIs shows a clear family-based clustering and separation into neurokine, pleiotrophin, FZD domain containing branches (*Figure 6G*). A node/edge network of PPI discovered similarly show two distinct clusters of interactions (*Figure 6H*; Clusters 1 and 2) that overlap with tissue expression clustering (*Figure 6F*).

SMOC1, a SPARC-related ligand overexpressed in brain tumors (*Brellier et al., 2011*), recovered BST2, IGSF23, and SMOC2 as the highest-ranking hits (*Figure 6I and J*). We observed binding of SMOC1 to SMOC2 by SPR (*Figure 6K*, blue sensogram). No binding was observed to IGSF23 or BST2 (*Figure 6K*; red sensograms). We also observed binding of SMOC1 to itself (*Figure 6L*; blue sensogram). SMOC2 is a matricellular protein which promotes matrix assembly and is involved in endothelial cell proliferation and migration, more recently SMOC2 variants were also reported to play a role in BMP signaling (*Long et al., 2021*).

## The scavenger receptor CD36 acts as a receptor for a broad range of secreted ligands

CD36, also known as SCARB3 or glycoprotein 4 (GPIV), is a multifunctional Type B scavenger receptor with two transmembrane domains and an ~410 aa ECD. CD36 is known to bind to many ligands (*Silverstein and Febbraio, 2009*). In our analysis of TM2+ library screens we identified CD36 as a top scoring hit in several screens: KRTDAP, LY6H, NRN1, NRN1L, VWC2L, and SCRG1 (*Figure 7A*). To examine potential binding of these secreted ligands to CD36 on the cell surface, fluorescent tetramers of LY6H, NRN1, VW2CL, and SCRG1 were incubated with 293F cells or 293F cells expressing full-length CD36 (*Figure 7B*, *Figure 7—figure supplement 1A*). We observed binding of LY6H, NRN1, VW2CL, and SCRG1 to CD36 by FACS. SPRC, which showed no enrichment of CD36 in its screen, served as a control and did in fact not bind to CD36 (*Figure 7B, C*). All CD36 enriched secreted ligands (LY6H, NRN1, NRN1L, VWC2L, SCRG1) except for KRTDAP show a strong correlation and cluster in brain tissue, which expresses CD36 mRNA only at low levels (*Figure 7D*). A PHA of all screens enriched for CD36 shows that VWC2L, LY6H, NRN1, NRN1L, and to some extent SCRG1 are more closely related to each other by sequence than KRTDAP (*Figure 7E*). Similarly, a multivariate analysis depicts a strong correlation of all secreted ligand with the exception of KRTDAP and CD36 itself (*Figure 7F*) in normal tissue. However, CD36 protein is expressed in brain microglia, and interestingly CD36-mediated debris uptake regulates brain inflammation in neurodegenerative disease models (*Dobri et al., 2021*; *Grajchen et al., 2020*).

While we also initially identified top scoring candidates for several other screens: MYOC, TICN1, OMG, and MDK, further analysis of their subcellular localization revealed that TMEM14A and REEP4 (indicated by dotted lines in the respective plots) are mainly localized in the ER or mitochondrial membrane compartment and not in the plasma membrane.

## Discussion

In an effort to accelerate the discovery of novel interactions between secreted ligands and the membrane proteome, we have developed a proteo-genomic high-throughput cell-based CRISPRa enrichment screening platform by employing customized, pooled cell surface receptor sgRNA libraries encoding transmembrane proteins in combination with MACS to enrich for receptor-expressing cells. We defined a list of 80 high-priority secreted ligands that are likely to have receptors that have not been previously identified. Twenty of these were successfully expressed, biotinylated, coupled to strepta-vidin magnetic beads, and used in our CRISPRa enrichment screening workflow. After three consecutive rounds of selections and deep sequencing of enriched cells, casTLE statistical analysis enabled high-confidence prediction and identification of interaction pairs from each dataset (*Figures 1 and 2*).

To validate the results of the screens, we used SPR and cell binding methods (CSS) to prove that the receptor candidates actually bound to the ligands used for screening. We then expanded the set of receptor candidates by taking advantage of homology (PHA), examining other members of the gene families identified in the initial screens for binding. Using these approaches, we identified 14 candidate receptors for 12 screened secreted ligands.

Interestingly, in some cases (OMG, TAFA-2), a single ligand bound to multiple receptors in the same family (*Figures 3 and 5*), while in other cases (CD36, RNF167), a single receptor bound to several unrelated ligands (*Figures 6 and 7*). Our results highlight the inability of standard high-throughput in vitro screening methods to identify genuine ECD interactions, because none of the hits from our screens were found in the aggregate of commonly used protein interaction databases (*Figure 2D*) and attests for the sensitivity and robustness of our screening strategy for secreted ligands. This was previously observed when the results of ELISA-based in vitro screens using multimeric ECD fusion proteins were compared with PPIs represented in these databases (*Martinez-Martin, 2017*; *Özkan et al., 2013*; *Ranaivoson et al., 2019*; *Söllner and Wright, 2009*; *Taouji et al., 2009*; *Verschueren et al., 2020*; *Wojtowicz et al., 2020*).

PPIs often occur between phylogenetically related proteins both within and between subfamilies and PPI modules with tight functional relationships similarly are more likely to be co-expressed. While tissue expression correlations can be used as a predictive tool for physical interactions, poor correlations do not necessarily suggest a lack of biological significance especially in the case of secreted ligands which are not restricted to the tissue origin of expression. In agreement with this hypothesis,

we observed that many of the protein pairs newly identified from the CRISPRa enrichment screens displayed tissue-dependent associations. In some cases, correlations were observed between secreted ligand and the receptor (GAS1, OSTN, OMG, TAFA-2), while in other cases correlations very strongly correlated between hit candidates (CD36).

Our results suggest that the cell-based screening strategy is primarily limited by two factors. First, fewer than one-third of the high-priority ligand candidates were expressed at high enough levels to allow purification for screening. If it were possible to express all of the 206 ligands we identified as targets, one might expect to be able to identify receptors for more than 100 of them. However, it is unclear how this might be accomplished. Second, while a positive selection strategy using CRISPRa has a higher dynamic range compared to loss-of-function screens, successful enrichment and recovery of a potential cell surface receptor PPI is dependent on the efficiency and level of CRISPRa-mediated expression of cell surface receptors. Also, many functional receptors are protein complexes and interactions might involve coreceptors or require chaperones that might not be present in K562 cells. In these cases, sgRNA-driven expression of single genes by CRISPRa would not be able to generate cells that make functional receptor complexes, unless the parental K562 cell line also made the other receptor complex components.

## New biology revealed by CRISPRa enrichment screens

The most striking finding from our screen was the identification of three new ligands for RPTPs, a diverse set of cell signaling receptors whose functions are less well understood than those of RTKs. OMG, a ligand expressed by oligodendrocytes and some neurons, binds to all three R3 RPTPs and to one of the R2B RPTPs, PTPRU (*Figure 3*). All of these RPTPs are expressed in the brain. Most interestingly, OSTN, a hormone that regulates bone and muscle growth in rodents but has acquired brain-specific expression and function in primates (*Ataman et al., 2016*), binds only to PTPRU (*Figure 3*). OSTN was reported to be a ligand for NPR-C, which regulates the levels of natriuretic peptides, but this is unlikely to explain its function in the primate brain. PTPRU, however, is expressed in the brain and is closely related to the three other R2B RPTPs, which are homophilic adhesion molecules that regulate cadherin-mediated cell adhesion. Moreover, PTPRU cannot mediate cell adhesion on its own, and is thought to lack phosphatase activity (*Hay et al., 2020*), so it may have a distinct function from the other R2B RPTPs.

GAS1, a regulator of Ret signaling involved in control of cell growth and a mediator of cell death (*Cabrera et al., 2006*), binds to PTPRA, an R4 RPTP that has no known ligand (*Figure 4*). PTPRA is a ubiquitously expressed signaling protein that both positively and negatively regulates tyrosine kinase signaling via dephosphorylation of Src family tyrosine kinases (*Mustelin and Hunter, 2002*). GAS1 may provide a link of PTPRA to RET RTK signaling complexes.

Furthermore, we discovered an unexpected association between TAFA-2, which is known as a neurokine (brain chemokine), and KIRs, which are inhibitory MHC class I receptors that are expressed only by NK cells (*Figure 5*). The TAFA2 gene is also expressed by T-regs and basophils, and TAFA-2 might be a soluble ligand that modulates NK signaling in response to MHC class I engagement. Furthermore, the newly discovered KIR3DL3-TAFA2 interaction may represent another intriguing way of modulating the KIR3DL3/TMIG2-HHLA2 signaling axis within the tumor microenvironment (*Li et al., 2022*; *Wei et al., 2021*).

Surprisingly, we also found several shared interactions between the PA-TM-RING E3 ligase RNF167, pleiotrophin, and neurokine family members (*Figure 6*). Most notably members of the PA-TM-E3 ligase family are RNF43 and ZNRF3 which both modulate the WNT signaling pathway by targeting FZD receptor protein homeostasis. Interestingly, both E3 ligases are regulated by the secreted WNT agonists R-spondin (RSPO) (*Clevers and Nusse, 2012*; *Janda et al., 2017*; *Zebisch and Jones, 2015*). Ubiquitin E3 ligases are notoriously challenging to study due to the highly transient, catalytic nature of the ubiquitylation cascade and the transmembrane PA-TM-RING E3 ligase family is no exception with the majority still being orphans and their functions unknown.

In summary, we implemented a proteo-genomic screening workflow, combining CRISPRa pooled cell surface libraries with a MACS enrichment strategy, in order to accelerate the identification of interactions between the secreted and membrane proteomes. We report new receptor-secreted ligand PPIs that are potentially involved in a wide variety of signaling processes. Implementation of cell-based screening strategies based on our approach might allow elucidation of receptor-ligand

relationships for many proteins that are currently orphans, and has the potential to identify novel therapeutically relevant targets and define new biological processes.

# Materials and methods

## Cell lines

Suspension cells were grown in plain bottom, vented flasks (Thermo Fisher Scientific), adherent cells were grown in T25 or T75 flasks (Thermo Fisher Scientific). Cells were maintained at 37°C and 5% $CO_2$. K562 (CCL-243; ATCC) cells were grown in RPMI supplemented with 10% fetal bovine serum (FBS), 1% GlutaMax, and 1% penicillin/streptomycin. HEK293T (CRL-3216; ATCC) and LentiX cells were maintained in DMEM supplemented with 10% FBS, 1% GlutaMax, and 1% penicillin/streptomycin. K562 SunTag-VP64 (CRISPRa) cell line was a gift from M Bassik. NKL cells including NKL-KIR2DL1 and KIR3DL1 expressing NKL cells were a gift from P Parham. HEK293F (R79007; Thermo Fisher Scientific) were grown in FreeStyle media (12338018; Thermo Fisher Scientific). Expi293F (A14528; Thermo Fisher Scientific) cells were grown in Expi293 Expression Medium (Thermo Fisher Scientific). Insect Hi5 cells (Tni; Expression Systems, 94–002S) were grown in ESF 921 media (Expression Systems) with a final concentration of 10 mg $l^{-1}$ of gentamicin sulfate (Thermo Fisher) at 27 °C and atmospheric $CO_2$. Cell lines tested negative for mycoplasma (MycoAlert Mycoplasma Detection kit, Lonza).

## Lentivirus production

HEK293T (LentiX) cells (female-derived kidney cell line) were grown in DMEM complete media (Thermo Fisher Scientific) supplemented with 10% FBS, 2 mM L-glutamine, 50 U/ml of penicillin and streptomycin, and used to package lentivirus using Fugene HD (Promega) in OptiMem (Thermo Fisher Scientific) as per the manufacturer's instructions. Third-generation packaging plasmids were used for the pooled sgRNA CSR libraries. After 72 hr, lentivirus containing media was harvested, filtered (0.45 µM pore, PVDF), and concentrated using PEG-it (SBI) according to the manufacturer's protocol or lentivirus containing media was used directly to infect the specified cell line.

## FACS staining

Cells were stained with the indicated antibodies at 1:100 dilution or tetramer at the indicated concentration for 30 min on ice in MACS staining buffer (Miltenyi). After incubation with fluorescent antibodies, cells were washed with MACS buffer and analyzed via flow cytometry on a Cytoflex (Beckman Coulter) instrument. Surface expression was quantified by FACS using the CytoFLEX equipped with a high-throughput sampler. Live cells were identified after gating on the basis of forward scatter and side scatter and propidium iodide (PI) negative staining. Data were analyzed using FlowJo 10.8.1 (BD). All assays were performed using independent biological replicates. The number of replicates (n) is indicated in the figure legends. Mean fluorescence intensity (MFI) was determined in FlowJo 10.8.1.

## Antibodies

Primary antibodies used in this study include anti-DYKDDDDK Tag (CST, D6W5B, # 15009), anti-CD36 (BioLegend, # 336206), anti-KIR3DL1 (BioLegend, # 312716), anti-anti-KIR3DL2 (R&D, # FAB2878A), anti-anti-KIR3DL3 (R&D, # FAB8919r), anti-KIR2DL2/L3 (BioLegend, # 312612), anti-CD122 (BioLegend, # 105912), anti-CD5 (BioLegend, # 364016), anti-CD25 (BioLegend, 302610), anti-CD272 (BioLegend, # 344510), anti-CD2 (BioLegend, # 300214), anti-CD28 (BioLegend, # 302912), anti-CD80 (BioLegend, # 305219), anti-CD45 (BioLegend, # 304012), anti-IL6ST (BioLegend, # 362006), anti-CD276 (BioLegend, # 351006), anti-CD47 (BioLegend, # 323124). These antibodies were used at 1:100 dilution in MACS staining buffer (Miltenyi).

## Curation of the human membrane and secreted proteome and selection of secreted bait proteins

Two lists of human membrane and secreted proteins were generated using the Human Protein Atlas (HPA) database (https://www.proteinatlas.org/humanproteome/tissue/secretome). Lists were checked for overlap and master lists were generated using the Human Protein Atlas majority decision-based method. Metadata for all master list proteins was extracted from UniProt https://www.uniprot.org and sequence information was validated by the SIgnalP-5.0 (http://www.cbs.dtu.dk/services/SignalP/)

(*Almagro Armenteros et al., 2019*), TMHMM-2.0 (https://services.healthtech.dtu.dk/service.php?
TMHMM-2.0), and PredGPI (http://gpcr.biocomp.unibo.it/predgpi/) (*Pierleoni et al., 2008*) prediction
servers. Following curation, canonical protein sequences were extracted from UniProt and compiled
for back translation and optimization by GeneArt/Life Sciences Technology for gene synthesis.

## Generation of secreted mammalian expression plasmids

Genes encoding curated SPOI were synthesized at GeneArt/Life Sciences Technologies and subcloned
into pD649-SPOI-AviTag-6xHis. Genes were subcloned in-frame with the endogenous or HA (influenza
hemagglutinin) signal peptide and downstream AviTag-6xHis modules via 5' NheI and 3' AscI sites.
A MaxiPrep of plasmid DNA was provided at 1 µg/ml in 20 mM Tris, pH 8.0. For complete plasmid
sequences of all 80 SPOI bait expression vectors, see *Figure 2—source data 3*. All plasmids (80) will
be made available through Addgene.

## Generation of expression plasmids for full-length proteins

Genes encoding full-length proteins were synthesized at GeneArt/Life Sciences Technologies and
subcloned into the pHR expression vector. Plasmids contain a kozak sequence, HA signal peptide, a
FLAG tag (to facilitate cell surface expression analysis), and the remaining full-length coding region of
the gene, followed by a stop codon.

## Production of purified proteins

Proteins were produced in Expi293F cells using transfection conditions following the manufacturer's
protocol. After harvesting of cell media, 1 M Tris, pH 8.0 was added to a final concentration of 20 mM.
Ni-NTA Agarose (Qiagen) was added to ~5% media volume. ×1 sterile PBS, pH 7.2 (Gibco) was added
to ~×3 media volume. The mixture was stirred overnight at 4°C. Ni-NTA agarose beads were collected
in a Buchner funnel and washed with ~300 ml protein wash buffer (20 mM HEPES, pH 7.2, 150 mM
NaCl, 20 mM imidazole). Beads were transferred to an Econo-Pak Chromatography column (Bio-Rad)
and protein was eluted in 15 ml of elution buffer (20 mM HEPES, pH 7.2, 150 mM NaCl, 200 mM imid-
azole). Proteins were concentrated using Amicon Ultracel filters (Millipore) and absorbance at 280 nm
was measured using a Nanodrop 2000 spectrophotometer (Thermo Fisher Scientific) to determine
protein concentration. A summary of the expression yields can be found in *Figure 2—source data 2*.

## Biotinylation and FPLC purification

Where indicated, proteins were biotinylated as described previously (*Özkan et al., 2013*). Briefly, up
to 10 mg of protein was incubated at 4°C overnight in ×2 Biomix A (0.5 M bicine buffer), ×2 Biomix B
(100 mM ATP, 100 mM MgOAc, 500 µM D-biotin), Bio200 (500 µM D-biotin) to a final concentration
of 20 µM, and 60–80 units BirA ligase in a final volume of 1 ml. Proteins were further purified by SEC
using an S200 Increase or a Superose S6 column (GE Healthcare), depending on protein size, on an
ÄKTA Pure FPLC (GE Healthcare), FPLC traces for purified proteins used for the CRISPRa enrichment
screens and SPR validation can be found in *Figure 2—figure supplement 2* and *Figure 2—figure
supplement 3*.

## CRISPRa enrichment screen

K562 cells stably expressing the sunCAS9 system carrying the pooled sgRNA CSR libraries (TM1;
TM2+) were expanded and 50 million cells per screen and library were harvested, washed three times
with cold MACS (Miltenyi) buffer, and resuspended in 2 ml MACS buffer in a sterile 5 ml Eppendorf
tube. Cells were then labeled with magnetic streptavidin microbeads complexed with biotinylated
bait protein (50 µl streptavidin microbeads; 1 µM biotinylated protein), mixed and incubated at 4°C
for 30 min (tumbling). After labeling, cells were washed twice with cold MACS buffer (300× *g*, 10 min),
resuspended in 1 ml MACS buffer and passed through a 40 µM cell strainer, to obtain a single-cell
suspension, directly onto the LS-Column (Miltenyi) for magnetic bead separation according to the
manufacturer's protocol. Briefly, after applying the labeled cells onto the LS-Column, unlabeled cells
pass through and are discarded while labeled cells are retained in the magnetic field, the LS-Column
is washed three times with 3 ml of ice-cold MACS buffer. For elution of positively selected cells, the
column is removed from the separator (magnet) and the magnetically labeled cells are flushed into a

15 ml Falcon tube with fresh media (RPMI complete), washed once, resuspended, and transferred to a T25 culture flask for expansion.

## Genomic DNA extraction, library amplification, and deep sequencing

Genomic DNA was isolated using QiaAmp DNA Blood Maxi or QiaAmp DNA mini kits (Qiagen) according to the manufacturer's instructions, genomic DNA was then amplified using Herculase II polymerase (Agilent) as described previously (*Deans et al., 2016*). To prepare the sgRNA sequencing library, the integrated sgRNA-encoding constructs were PCR amplified using Agilent Herculase II Fusion DNA Polymerase, followed by a second PCR amplification introducing sample-specific Illumina index barcodes and adapters for deep sequencing. Deep sequencing was performed using the MiSeq Reagent Kit v2 (300 cycles; Illumina) employing a custom sequencing oligo according to the manufacturer's instructions:

(5′-GCCACTTTTTCAAGTTGATAACGGACTAGCCTTATTTAAACTTGCTATGCTGTTTCCAGCTTAG CTCTTAAAC-3′).

## Cell surface binding assay with streptavidin tetramerized secreted ligand

To examine PPIs at the cell surface, we performed cell surface protein binding assays using K562 or HEK293F cells. K562 cells were used for pre-evaluation of potential base line binding for all secreted proteins used in the CRISPRa screening workflow. HEK293F cells were transfected using Fugene6 according to the manufacturer's protocol (Promega) with expression plasmids encoding full-length proteins containing an N-terminal tag (FLAG). Two days following transfection, cultures were harvested, cells were spun down for 4 min at 1600 rpm (~400× $g$), washed twice with cold MACS buffer (Miltenyi) and resuspended to a final density of ~3 × $10^6$ cells/ml. To generate tetramerized secreted ligands to test for binding to cells expressing full-length proteins, FPLC-purified biotinylated proteins (see above) were incubated with streptavidin tetramers conjugated to Alexa647 Fluor (SA-647) (Thermo Fisher Scientific) at a 4:1 molar ratio on ice for at least 15 min. To assess cell surface expression of full-length or ECD displayed proteins, 1:200 mouse anti-FLAG-647 (CST) or anti-HA antibody (CST) staining of cells was also performed in parallel where indicated. Approximately 150,000 cells were incubated with Protein:SA-647 complexes or antibody in a final volume of 100 µl in 96-well round-bottom plates (Corning) for 1 hr at 4°C protected from light. Following incubation, cells were washed two times with 200 µl cold MACS buffer and resuspended in 200 µl cold MACS buffer with 1:3000 PI (Thermo Fisher Scientific). Immunofluorescence staining was analyzed using a Cytoflex (Beckman Coulter), and data were collected for 20,000 cells. Data were analyzed using FlowJo v10.4.2 software. All data report MFI. Concentration-dependent binding of Protein:SA-647 to full-length receptor expressing, but not mock control cells, was deemed indicative of cell surface binding.

## SPR experiments

SPR experiments were performed using a Biacore T100 instrument (GE Healthcare). FPLC-purified biotinylated proteins (ligands) in HBS-P+ Buffer (GE Healthcare) were captured on a Streptavidin (SA) Series S Sensor Chip (GE Healthcare). Chip capture was performed in HBS-P+ buffer (GE Healthcare) to aim for ~100–200 ligand response units (RU). Flow cell 1 was left empty to use this flow cell as a reference flow cell for online subtraction of bulk solution refractive index and for evaluation of non-specific binding of analyte to the chip surface using Biacore T100 Control Software (v3.2) (GE Healthcare). FPLC-purified non-biotinylated protein was used as analyte. Analytes were run in HBS-P+ buffer using twofold increasing protein concentrations to generate a series of sensorgrams. Binding parameters were either determined based on a 1:1 Langmuir model or at equilibrium using the accompanying Biacore T100 evaluation software. A table of all SPR conditions for each ligand-analyte pair tested including concentration range of twofold analyte dilutions, injection rate, injection and dissociation times, regeneration conditions, and meta data can be found in *Figure 2—source data 5*.

## Data analysis

Deep sequencing results for each round of selection were analyzed using the casTLE statistical framework (*Morgens et al., 2016*). Briefly, casTLE compares each set of gene-targeting guides to the negative controls, using both safe-targeting and non-targeting controls and selecting the most likely

maximum effect size (casTLE-Effect). A pValue is then generated, representing the significance of this maximum effect by permuting the results (n=10,000 permutation). Screening results of the final round of enrichment for each of the 20 secreted proteins were subjected to a first-pass filter using the following cut-offs: casTLE-Effect > 2, casTLE-Score > 2, pValue < 0.05. Next, hits outside the secreted and membrane proteome and common false positive contaminants were removed. To predict high-confidence interaction pairs from each dataset, a custom score was then computed for each potential interaction pair by combining all three metrics into one ESP score: (casTLE-Effect + casTLE Score)/ pValue. To integrate data analysis and visualization, we used the combined ESP score to rank sort interaction pairs for every screen and created trajectory plots to ensure positive enrichment over the three consecutive rounds of selection for the predicted hits and allows elimination of false positive candidates.

## Database integration

Interaction datasets were downloaded from BioGRID (https://thebiogrid.org, 4.4.210, physical inter-actions), Bioplex (https://bioplex.hms.harvard.edu, Hek293 v3.0 and HCT116 v1.0) and STRING (https://string-db.org/; v11.5; physical dataset; interaction score >0.4). To calculate the overlap between all obtained datasets and out own study, interactions were restricted to physical interactions reported and a Venn diagram was visualized (*Heberle et al., 2015*). PHA was performed to generate phylogenetic trees from multiple sequence alignments (MSA) of amino acid sequences of secreted or ECD sequences of transmembrane cell surface receptors (https://www.uniprot.org/). Briefly, MSA was performed using ClustalOmega (https://www.ebi.ac.uk/Tools/msa/clustalo/) and alignments results were submitted to calculate phylogenetic tree parameters (https://www.ebi.ac.uk/Tools/phylogeny/simplephylogeny/) which were visualized by Interactive Tree of Life (iTOL; https://itol.embl.de/) (*Letunic and Bork, 2021*). Tissue expression datasets, normal tissue, and TCGA (The Cancer Genome Atlas) datasets were downloaded from The Human Protein Atlas (https://www.proteinatlas.org; v21.1). TCGA cancer tissue RNA-seq data was obtained from 17 cancer types representing 21 cancer subtypes and were processed as median FPKM (number fragments per kilobase of exon per million reads) and visualized as a hierarchical clustering heatmap using JMP Pro (v16). Unsupervised hierarchical clustering of normalized mRNA gene expression by tissue was performed with Ward linkage and correlation distance were plotted as heatmaps using JMP Pro (v16). Tissue expression correlation analysis of normalized mRNA gene expression for candidate genes was performed using multivariate analysis tool in JMP Pro (v16) and expression correlation results were visualized as heatmaps where intense red color indicates a strong positive correlation and intense blue color indicates a strong negative correlation.

## Acknowledgements

The authors were supported by The Howard Hughes Medical Institute (KCG), the G Harold and Leila Y Mathers Charitable Foundation (KCG), and NIH 1RO1-GM150125 (KCG and KZ) .

## Additional information

### Funding

| Funder | Grant reference number | Author |
|--------|------------------------|--------|
| Howard Hughes Medical Institute | | K Christopher Garcia |
| G Harold and Leila Y Mathers Charitable Foundation | | K Christopher Garcia |
| National Institute of General Medical Sciences | 1 RO1 GM150125 | K Christopher Garcia Kai Zinn |

The funders had no role in study design, data collection and interpretation, or the decision to submit the work for publication.

## Author contributions
Dirk H Siepe, Conceptualization, Data curation, Formal analysis, Validation, Investigation, Visualization, Methodology, Writing - original draft, Writing – review and editing; Lukas T Henneberg, Investigation; Steven C Wilson, Validation; Gaelen T Hess, Michael C Bassik, Methodology; Kai Zinn, Data curation, Writing - original draft, Writing – review and editing; K Christopher Garcia, Conceptualization, Supervision, Funding acquisition, Investigation, Methodology, Project administration, Writing – review and editing

## Author ORCIDs
Dirk H Siepe ⓘ http://orcid.org/0000-0002-0009-8023
Lukas T Henneberg ⓘ http://orcid.org/0000-0003-3477-4541
Michael C Bassik ⓘ http://orcid.org/0000-0001-5185-8427
Kai Zinn ⓘ http://orcid.org/0000-0002-6706-5605
K Christopher Garcia ⓘ http://orcid.org/0000-0001-9273-0278

## Decision letter and Author response
Decision letter https://doi.org/10.7554/eLife.81398.sa1
Author response https://doi.org/10.7554/eLife.81398.sa2

---

# Additional files

## Supplementary files
• MDAR checklist

## Data availability
All data generated or analyzed during this study are included in the manuscript and supporting files.

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
