## [Editor Report]

This paper reports the development and application of a proteo-genomic screening platform to identify protein-protein interactions between secreted proteins and their cell surface receptors. The authors use a CRISPRa-based approach to overexpress membrane proteins in cells and then use magnetic cell sorting to identify receptors that bind candidate ligands. This approach led to the identification of several novel interaction pairs that were then validated biochemically, including receptor tyrosine phosphatase ligands and other interactions with implications for immune system function. The work is likely to be relevant to a wide variety of fields including biochemistry and signal transduction research.

---

## [Decision Letter]

**Decision letter after peer review:**

Thank you for submitting your article "Identification of orphan ligand-receptor relationships using a cell-based CRISPRa enrichment screening platform" for consideration by *eLife*. Your article has been reviewed by 3 peer reviewers, and the evaluation has been overseen by a Reviewing Editor and Jonathan Cooper as the Senior Editor. The reviewers have opted to remain anonymous.

Essential revisions:

The reviewers were uniformly enthusiastic about the work but raised a variety of small points for clarification which should be addressed in a revised manuscript. Some additional experiments were suggested (see reviewer 3 point 10) but reviewers agreed that this new data is not strictly essential, although it could strengthen the manuscript if it were to be added. We leave this latter decision to the authors' discretion.

*Reviewer #1 (Recommendations for the authors):*

This manuscript is well-written and made the workflow for a complex screen easy to follow for non-specialists. The data is rigorous and includes all appropriate controls. I have only a few comments/critiques:

1) It was surprising that CD36 was identified as a receptor for 4 out of 20 ligands tested on the screen. Can the authors comment on the potential of scavenger receptors or other promiscuous binders to outcompete other top hits? Did CD36 score highly for the other 16 ligands screened?

2) The authors should speculate on why TAFA-2 would selectively bind to D0-containing KIRs. Are KIR subtypes expressed on certain subsets of tissues or cells such that this makes biological sense?

3) The identification of PTPRU as the bona fide receptor for OSTN was notable in that it was not the top hit and required trajectory analysis of the three rounds of selection. On a high-throughput basis, deep sequencing of this many samples could become quite costly. The authors should comment on whether they recommend trajectory analysis as a routine operation in hit identification, or whether it could it be possible to circumvent this by performing additional rounds of selection.

*Reviewer #3 (Recommendations for the authors):*

1) The authors should more clearly state their use of the Chong et al., 2018 method paper as the starting point for this work. That paper describes a CRISPR activation screen that included sgRNAs targeting the whole human secretome in HEK 293 cells, which is then used to query soluble biotinylated GPCRs to identify novel interactions. The current authors should elaborate on the similarities and any specific differences between their approach and the Chong paper. The current treatment appears to give the Chong paper short shrift and reduces the purported novelty of the work.

2) The authors could go into more detail regarding their selection of the 80 soluble targets for expression. Contrary to statements made, many of the targets are directly related to one another (Ex: 8/80 are angiopoietin-related proteins). Also, over half of the targets screened are primarily found in the brain, so the tissue distribution doesn't seem that broad either. The text mentions that disease associations were taken into account; however, there are no data supporting this claim.

3) Have any of the 60 query proteins that failed to express been successfully expressed using other recombinant protein systems? Did the authors consider any rescue strategies for improving recombinant expression success rates?

4) Why were K562 cells chosen for this study as opposed to using HEK 293 cells that were used in other published CRISPRa screens? Are there observed differences in the CRISPR activation efficiency between these cell lines? HEK 293 cells are well established as a model system for high protein expression, especially for secreted and membrane proteins, suggesting it might be a better choice. Were other candidate cell lines tested?

5) What fraction of the library genes are upregulated to significant levels? In the assessment of the small TM1 screen, it is claimed that all 10 of the genes showed elevated cell surface expression. However, looking at the FACS plots in Figure S1 (B), this does not seem to be accurate. CD5 and IL6ST showed no change and others exhibited very modest increases in expression. Is this level of expression sufficient to identify known ligands of these proteins (e.g., weak binders)? This information would help to assess the robustness of this approach and how consistently it identifies targets with a range of different expression levels and different binding affinities.

6) The IL-2/CD25 test demonstrates that the TM1 screen can efficiently identify a high affinity (Kd ~10-8) interaction with high confidence. However, several of the interactions identified have affinities in the range of 10-6 or lower. The authors should consider running control screens against the full TM1 and TM2 libraries using soluble queries (CD80, CD200, etc.) that bind known receptors with μM affinities. These controls would help evaluate how well this approach identifies established, biologically relevant, low-affinity interactions.

7) In some of the screens (e.g., OMG, GAS1) other genes also show high ESP scores. Were any of the second or third highest scoring "hits" characterized further for potential enrichment in subsequent rounds of screening? The scale of the ESP plots also changes significantly depending on the screen. In the original Chong et al., paper, only one round of enrichment was used and all of the ERA plots are set to the same scale. They used a larger library and clearly demonstrated to ability to identify known weak interactions. Do multiple rounds of enrichment increase the potential for amplifying screening artifacts/nonspecific/nonrelevant interactions? The scoring scheme is also slightly different from Chong et al., making it difficult to directly compare the two studies.

8) In the OSTN screen no enrichment of PTPRU was observed until 3 rounds of selection, even though the measured affinity is higher than that determined for the OMG/PTPRU interaction. Is there an explanation as to why that was the case? The overall level of enrichment also seems lower. There is also a target gene (red dot) in Figure 3G between PTPRJ and PTPRU. Is there a strong correlation between rounds of enrichment and Kd? What is the identity of that target and what was its trajectory during the 3 rounds of enrichment?

9) For the SPR experiments, many of the runs required high analyte protein concentrations (80uM for TAFA2, 60uM for OMG). Are the authors confident that the proteins used for SPR were not forming soluble aggregates at the highest analyte concentrations used for binding? Soluble aggregates notoriously bind nonspecifically and can mimic low-affinity interactions.

10) For the cell binding experiments, negative control tetramer titrations should also be performed using a tetramerized protein not expected to bind. Also, the cell binding titrations do not saturate and the binding only appears statistically significant from controls at the highest concentrations used. Do the reciprocal cell experiments also show binding? These data would benefit from the inclusion of representative FACS plots and a description of the gating strategy used.

11) There are target genes (red dots) with relatively high ERA scores in the TM2+ screens that are not CD36 (For example in the NRN1L, TICN1, SCRG1, and MK screens). What are these targets and was binding further characterized for any of them? As CD36 seems to act as a scavenger receptor, are there sequence similarities or other shared characteristics between the receptors CD36 interacts with?

---

## [Author Response]

Reviewer #1 (Recommendations for the authors):This manuscript is well-written and made the workflow for a complex screen easy to follow for non-specialists. The data is rigorous and includes all appropriate controls. I have only a few comments/critiques:

We are very happy about this enthusiastic comment that the manuscript is “well-written and made the workflow for a complex screen easy to follow for non-specialists” and “data is rigorous and includes all appropriate controls”. We are delighted that this reviewer described our screening method as of “immediate scientific interest given the growing need to identify receptors for orphan ligands”, “powerful new tool for ligand deorphanization in the extracellular space”. We address his/her specific criticisms below.

(1) It was surprising that CD36 was identified as a receptor for 4 out of 20 ligands tested on the screen. Can the authors comment on the potential of scavenger receptors or other promiscuous binders to outcompete other top hits? Did CD36 score highly for the other 16 ligands screened?

Regarding the reviewer’s question about the CD36, the scavenger receptor CD36 was indeed reported as a promiscuous binder with many targets involved numerous signaling pathways (Silverstein and Febbraio, 2009). Since the screening platform is based on a bead based enrichment workflow using MACS (Magnetic-Activated Cell Sorting) it is of course always a possibility that one early round binder might outcompete other low affinity binders.

CD36 only specifically enriched for 6 secreted ligands: LY6H, KRTDAP, NRN1, NRN1L, VWC2L and SCRG1. We tested 4 secreted ligands (LY6H, NRN1, VWC2L, SCRG1) against CD36 with 1 screen (SPRC) that did not show any enrichment, that was used as a negative control for the cell surface tetramer staining in addition to non CD36 transduced cells.

(2) The authors should speculate on why TAFA-2 would selectively bind to D0-containing KIRs. Are KIR subtypes expressed on certain subsets of tissues or cells such that this makes biological sense?

We appreciate the reviewer’s interest in the newly discovery interaction between TAFA2 and certain members of the D0 domain containing inhibitory KIR receptor family. We share the enthusiasm of this interaction in the greater context of HHLA2 and HLA and added HHLA2 as a point of interest in the Result and Discussion section.

(3) The identification of PTPRU as the bona fide receptor for OSTN was notable in that it was not the top hit and required trajectory analysis of the three rounds of selection. On a high-throughput basis, deep sequencing of this many samples could become quite costly. The authors should comment on whether they recommend trajectory analysis as a routine operation in hit identification, or whether it could it be possible to circumvent this by performing additional rounds of selection.

We agree with the reviewer’s comment and would like to point out that we used the trajectory analysis as part of our analysis workflow for our screens and further used the enrichment plots to predict potential family related interactions (Figure 1I, 3B, 3D, 3H, 4B, 4C, 5B, 6I, 6J) which we then followed up on. For future screens, a trajectory score could be considered as a useful metric. In our experience with this study and other screens, additional rounds of selections did not improve beyond three rounds of selections, at this point the CRISPRa libraries were usually depleted, and further screening only resulted in increased false positive binders.

Reviewer #3 (Recommendations for the authors):(1) The authors should more clearly state their use of the Chong et al., 2018 method paper as the starting point for this work. That paper describes a CRISPR activation screen that included sgRNAs targeting the whole human secretome in HEK 293 cells, which is then used to query soluble biotinylated GPCRs to identify novel interactions. The current authors should elaborate on the similarities and any specific differences between their approach and the Chong paper. The current treatment appears to give the Chong paper short shrift and reduces the purported novelty of the work.

We appreciate the reviewer’s comments. However, we would like to point out that we did not use the MS2-p65-HSF1 transcriptional activator platform that was described in Gavin Wright’s elegant study to screen interactions between cell surface receptors using receptor ECDs by FACS sorting (Chong et al., 2018). In the study presented here, we focused on interactions between the secreted and membrane proteome and used the SunTag system developed in Ron Vale’s lab (Tanenbaum et al., 2014) and was then adapted by Jonathan Weissman’s group for CRISPRa termed sunCas9 (Gilbert et al., 2014; Horlbeck et al., 2016). Since we planned on screening a large number of secreted ligands, we developed a MACS based enrichment workflow to screen many ligands in parallel without special instrumentation.

A large number of different CRISPRa transcriptional activator systems are currently in use and a direct comparison of different 2^nd^ generation systems showed that VPR, SunTag, and SAM approaches perform equally effective across a number of different cell lines (Chavez et al., 2016; Kampmann, 2018). To better describe the rationale, we refined the text sections in the main text accordingly:

“To address these issues, several groups have successfully developed cell-based screens for phenotypical screens (Bassik et al., 2013; Han et al., 2020; Kamber et al., 2021), uncovering signaling cascades (Breslow et al., 2018; Wisnovsky et al., 2021) or to study interactions between cell surface receptors using receptor ECDs (extracellular domains) (Chong et al., 2018), that take advantage of CRISPR technology (Cong et al., 2013; Jinek et al., 2013; Mali et al., 2013). In CRISPR activation (CRISPRa) screens such as the one described here, gene expression is induced by targeting transcriptional activators (Chavez et al., 2016; Chong et al., 2018; Tycko et al., 2017) to their control elements using sgRNAs (Kampmann, 2018; Morgens et al., 2016; Tanenbaum et al., 2014).

“A large variety of different 2nd generation dCas9 activator (CRISPRa) systems are currently in use. A recent study directly compared a large number of published Cas9 activators systems and found the VPR, SunTag, and SAM approaches perform equally effective across a number of different cell lines and target genes (Chavez et al., 2016; Kampmann, 2018b). The SunTag system was initially developed in K562 cells (myeloid leukemia cell line K562) (Tanenbaum et al., 2014b), a highly characterized (ENCODE, The Protein Atlas) and easy to handle suspension cell line that is widely used for CRISPR screens and highly suitable for MACS (Magnetic-Activated Cell Sorting) based applications.”

(2) The authors could go into more detail regarding their selection of the 80 soluble targets for expression. Contrary to statements made, many of the targets are directly related to one another (Ex: 8/80 are angiopoietin-related proteins). Also, over half of the targets screened are primarily found in the brain, so the tissue distribution doesn't seem that broad either. The text mentions that disease associations were taken into account; however, there are no data supporting this claim.

We thank the reviewer for his/her comment. Details with regard to the selection of the 80 secreted targets can be found in the Result section “Selection and production of secreted proteins for CRISPRa screening.”, where we outlined the selection workflow. We agree with the reviewer’s comment with regard to the tissue mRNA expression distribution and refined this section accordingly to address this point:

“From these 206, we ultimately selected a total of 80 high priority targets (one per gene; we did not consider isoforms generated through alternative splicing). These had a wide range of tissue expression with many of the chosen secreted ligands being expressed in brain tissue (Figure 2A) covering a broad range of molecular function and processes (Figure 2—figure supplement 1A, B).”

In addition, we added two panels to Figure 2 that include cancer related mRNA expression data (TCGA) (Figure 2B) and gene disease association data (GDA) (Figure 2C) for the 20 candidates that were ultimately used in the screening workflow. A GDA related cytoscape network was visualized in Figure 2—figure supplement 4A and the full GDA data for all of the 20 candidates was included in Figure 2-source data 3 and a fully annotated node/edge GDA network in cytoscape (Figure 2-source data 3.cys). To reflect the additional panels we refined the text section accordingly:

“The 20 high priority targets show a wide spectrum of tissue expression in normal, healthy tissue (Figure 2A) and a broad range of expression in cancer (Figure 2B) with several candidates enriched in brain tissue and clustering in Glioma (GBM, brain tissue and in various locations in the nervous system, including the brain stem and spinal column). In addition, the majority of targets show a strong gene disease association (GDA) with many reported diseases (Figure 2C, Figure 2—figure supplement 4, Figure 2-source data 3).”

(3) Have any of the 60 query proteins that failed to express been successfully expressed using other recombinant protein systems? Did the authors consider any rescue strategies for improving recombinant expression success rates?

We appreciate the reviewer’s question with regard to rescuing some of the secreted ligands by using an alternative expression system. We did in fact explore whether insect expression could serve as viable alternative expression strategy to rescue some of the secreted ligands that failed to express in the Expi293F expression system. To test this strategy we selected a total of 12 candidates that were subcloned for expression in insect cells (HI5): 10 candidates that showed no expression in the Expi293F expression system (PODN_HUMAN ,NOE1_HUMAN ,MYOC_HUMAN ,NPTX1_HUMAN ,ITGBL_HUMAN ,FGFP2_HUMAN ,CCBE1_HUMAN ,ANGL7_HUMAN ,ANGL3_HUMAN ,ANGL1_HUMAN) and an additional 2 candidates that were previously successfully expressed in Expi293F (CC134_HUMAN; SPRC_HUMAN). While the 2 (CC134_HUMAN; SPRC_HUMAN) control candidates showed some low level of expression in Hi5, none of the other 10 candidates showed any positive expression in insect cells. We included the alternative approach in the Results section and updated Figure 2-source data 2 to include the expression data for Expi293F and insect expression data accordingly.

(4) Why were K562 cells chosen for this study as opposed to using HEK 293 cells that were used in other published CRISPRa screens? Are there observed differences in the CRISPR activation efficiency between these cell lines? HEK 293 cells are well established as a model system for high protein expression, especially for secreted and membrane proteins, suggesting it might be a better choice. Were other candidate cell lines tested?

A variety of different CRISPRa systems are currently in use and a recent study directly compared a large number of published Cas9 activators systems and found the VPR, SunTag, and SAM approaches to perform equally effective across a number of different cell lines and target genes (Chavez et al., 2016; Kampmann, 2018). The SunTag system was initially developed in the myeloid leukemia cell line K562 (Tanenbaum et al., 2014). K562 cells are a highly characterized (ENCODE, The Protein Atlas) and easy to handle suspension cell line that is widely used for CRISPR screens and highly suitable for MACS (Magnetic-Activated Cell Sorting) based applications like the one used here.

(5) What fraction of the library genes are upregulated to significant levels? In the assessment of the small TM1 screen, it is claimed that all 10 of the genes showed elevated cell surface expression. However, looking at the FACS plots in Figure S1 (B), this does not seem to be accurate. CD5 and IL6ST showed no change and others exhibited very modest increases in expression. Is this level of expression sufficient to identify known ligands of these proteins (e.g., weak binders)? This information would help to assess the robustness of this approach and how consistently it identifies targets with a range of different expression levels and different binding affinities.

We thank the reviewer for his/her comment, and changed the text accordingly to reflect that the majority (8/10) targets show elevated expression. In addition, we added the anti-CD122 antibody staining panel and an Isotype control staining panel to Figure 1—figure supplement 1B and added the missing color legend.

(6) The IL-2/CD25 test demonstrates that the TM1 screen can efficiently identify a high affinity (Kd ~10-8) interaction with high confidence. However, several of the interactions identified have affinities in the range of 10-6 or lower. The authors should consider running control screens against the full TM1 and TM2 libraries using soluble queries (CD80, CD200, etc.) that bind known receptors with μM affinities. These controls would help evaluate how well this approach identifies established, biologically relevant, low-affinity interactions.

We appreciate the reviewers’ comment. Since CD80 or CD200 are members of the single transmembrane receptor superfamily and not secreted ligands, we felt it would be more in the general reader’s interest to see a benchmark screen with a real secreted ligand instead of receptor ECD (extracellular domain) based benchmarking of the libraries. However, we agree with the reviewer that in addition to screening orphan secreted ligands, our screening workflow could easily be adopted to screen for orphan transmembrane receptor protein-protein interactions using their respective receptor ECD’s (extracellular domains) including CD80 or CD200 among others.

(7) In some of the screens (e.g., OMG, GAS1) other genes also show high ESP scores. Were any of the second or third highest scoring "hits" characterized further for potential enrichment in subsequent rounds of screening? The scale of the ESP plots also changes significantly depending on the screen. In the original Chong et al., paper, only one round of enrichment was used and all of the ERA plots are set to the same scale. They used a larger library and clearly demonstrated to ability to identify known weak interactions. Do multiple rounds of enrichment increase the potential for amplifying screening artifacts/nonspecific/nonrelevant interactions? The scoring scheme is also slightly different from Chong et al., making it difficult to directly compare the two studies.

We thank the reviewer for the comments and agree that unless both studies would employ the exact same baits, dCas9 transactivation system, library design, gRNA sequences, cell line, screening workflow and statistical analysis, a direct side by side comparison of screening results and metrics from both strategies is very difficult. Regarding plots with either linear or log10 scale ESP, we tried to visualize the results of the casTLE statistical analysis as best as possible. In cases where we compared several candidate genes in trajectory plots, the scale was only changes to improve visualization for the reader.

In several screens presented here, other candidates that showed enrichment or where the result of a phylogenetic homology analysis were tested. In the GAS1 screen we tested interactions of PTPRA, PTPRU and PTPRJ by SPR. For the OMG and OSTN screen, we tested a panel of 8 PTPR sub family members by SPR. For the TAFA-2 screen, the top 3 tanking hits (KIR3DL1, KIR3DL3 and RNF167) were tested. For the PTN, MK, TAFA-2 and TAFA-5 screen several shared hits were tested. Additional SPR results were included in the supporting Supplemental figures (Figure 3—figure supplement 1A+C, Figure 4—figure supplement 1A+B, Figure 5—figure supplement 1A+B, Figure 6—figure supplement 1A+B).

(8) In the OSTN screen no enrichment of PTPRU was observed until 3 rounds of selection, even though the measured affinity is higher than that determined for the OMG/PTPRU interaction. Is there an explanation as to why that was the case? The overall level of enrichment also seems lower. There is also a target gene (red dot) in Figure 3G between PTPRJ and PTPRU. Is there a strong correlation between rounds of enrichment and Kd? What is the identity of that target and what was its trajectory during the 3 rounds of enrichment?

Since the screening platform is based on a bead based enrichment workflow using MACS (Magnetic-Activated Cell Sorting) we have the advantage to follow the enrichment of a candidate over the rounds of selections after deep sequencing and analysis for the different rounds. However, multiple factors might contribute to levels of enrichment e.g. level of activation, gRNA usage, fitness, number of binders, among others and might not reflect the actual affinity of a ligand to it’s receptor or molecular function. In cases where several binders where observed they might compete with each other during the enrichment workflow in earlier rounds while in other cases where there is only one binder, it outcompetes non-binders in earlier rounds of the selection. In general, we did not observe a clear correlation between enrichment and measured affinity. Figures were refined to include annotations for additional hits.

(9) For the SPR experiments, many of the runs required high analyte protein concentrations (80uM for TAFA2, 60uM for OMG). Are the authors confident that the proteins used for SPR were not forming soluble aggregates at the highest analyte concentrations used for binding? Soluble aggregates notoriously bind nonspecifically and can mimic low-affinity interactions.

To detect interactions in the μm affinity range as observed for some of the hits a 10x concentration over Kd is necessary for SPR. It is possible that soluble aggregates could bind, and we cannot be sure that these do not exist in our preparations. However, as the reviewer might appreciate, we included sensograms of non binding candidates for the indicated screens in the supplemental to show specificity of the positive binders (Figure 3—figure supplement 1A+C, Figure 4—figure supplement 1A+B, Figure 5—figure supplement 1A+B, Figure 6—figure supplement 1A+B). Furthermore, to validate the results, SPR is backed up by cell surface staining, which should not be subject to artifacts produced by soluble aggregates.

(10) For the cell binding experiments, negative control tetramer titrations should also be performed using a tetramerized protein not expected to bind. Also, the cell binding titrations do not saturate and the binding only appears statistically significant from controls at the highest concentrations used. Do the reciprocal cell experiments also show binding? These data would benefit from the inclusion of representative FACS plots and a description of the gating strategy used.

We appreciate the reviewers’ question and would like to direct the reviewer’s attention to Figure 7B/C, where we used SA647:SPRC tetramers to show that a screen not enriched for CD36 also shows no binding to CD36 expressing cells. In general, cell surface staining assays were performed in transduced and non-transduced cells and SA647:Biotin was used as a unrelated negative control. In addition we included sensograms of non binding candidates for the indicated screens in the supplemental to show specificity of the positive binders (see also response 9).

Reciprocal binding assays were considered but due to the fact that the baits are secreted ligands, they would be secreted into the media and detection by tetramer cell surface staining rendered unfeasible.

(11) There are target genes (red dots) with relatively high ERA scores in the TM2+ screens that are not CD36 (For example in the NRN1L, TICN1, SCRG1, and MK screens). What are these targets and was binding further characterized for any of them? As CD36 seems to act as a scavenger receptor, are there sequence similarities or other shared characteristics between the receptors CD36 interacts with?

We thank the reviewer for observations and questions, we refined Figure 7A accordingly by adding annotations for the following screens: LY6H, NRN1L, MYOC, TICN1, SCRG1, OMG and MK. As the reviewer will appreciate the 2^nd^ top scoring hit for the aforementioned screens generally scored thousands of ESP units lower than the top hit e.g., in the LY6H screen CD36 scored at ESP=441.000 ESP while the 2^nd^ best hit CMTM5 scored at ESP: 81.500. The fold change difference between the 1^st^ (CD36) and 2^nd^ highest scoring hit generally ranges between 2 to 10 fold changes, we therefore focused our analysis on the top scoring ESP candidate to predict high-confidence interaction pairs from each dataset. In addition, the top scoring candidate for TICN1, OMG and MK was identified asTMEM14A. While TMEM14A was annotated as multi-pass membrane protein with 3 transmembrane helices, it appears to be mainly localized to ER and mitochondrial membranes and was therefore not further considered. We added the annotations and changed the text section accordingly.

Furthermore, to explore potential functional relationships between CD36 as the top scoring hit and its baits, we added a phylogenetic homology analysis calculated from a multiple sequence alignment of the CD36 enriched screens (Figure 7E) and updated the text section accordingly:

“A phylogenetic homology analysis of all screens enriched for CD36 shows that VWC2L, LY6H, NRN1, NRN1L and to some extend SCRG1 are mor closely related to each other by sequence than KRTDAP (Figure 7E).”

We would also like to direct the reviewer to the result section “The Scavenger Receptor CD36 acts as a receptor for a broad range of secreted ligands.”, where we performed a hierarchical clustering analysis (Figure 7D) and a multivariate correlation analysis (Figure 7E) as indicated to provide potential gene expression connectivity:

“Similarly, a multivariate analysis depicts a strong correlation of all secreted ligand with the exception of KRTDAP and CD36 itself (Figure 7F) in normal tissue. However, CD36 protein is expressed in brain microglia, and interestingly CD36-mediated debris uptake regulates brain inflammation in neurodegenerative disease models (Dobri et al., 2021; Grajchen et al., 2020).”

References

Chong ZS, Ohnishi S, Yusa K, Wright GJ. 2018. Pooled extracellular receptor-ligand interaction screening using CRISPR activation. Genome Biol. doi:10.1186/s13059-018-1581-3

Gilbert LA, Horlbeck MA, Adamson B, Villalta JE, Chen Y, Whitehead EH, Guimaraes C, Panning B, Ploegh HL, Bassik MC, Qi LS, Kampmann M, Weissman JS. 2014. Genome-Scale CRISPR-Mediated Control of Gene Repression and Activation. Cell 159:647–661. doi:10.1016/J.CELL.2014.09.029

Horlbeck MA, Gilbert LA, Villalta JE, Adamson B. 2016. Compact and highly active next-generation libraries for CRISPR-mediated gene repression and activation. *eLife*.

Silverstein RL, Febbraio M. 2009. CD36, a Scavenger Receptor Involved in Immunity, Metabolism, Angiogenesis, and Behavior. Sci Signal 2:re3. doi:10.1126/SCISIGNAL.272RE3

Tanenbaum ME, Gilbert LA, Qi LS, Weissman JS, Vale RD. 2014. A Protein-Tagging System for Signal Amplification in Gene Expression and Fluorescence Imaging. Cell 159:635–646.